# Comprehensive machine-learning survival framework develops a consensus model in large-scale multicenter cohorts for pancreatic cancer

Libo Wang[1,2,3†], Zaoqu Liu[4*†], Ruopeng Liang[1,2,3†], Weijie Wang[1,2,3], Rongtao Zhu[1,2,3], Jian Li[1,2,3], Zhe Xing[5], Siyuan Weng[4], Xinwei Han[4*], Yu-ling Sun[1,2,3*]

[1]Department of Hepatobiliary and Pancreatic Surgery, The First Affiliated Hospital of Zhengzhou University, Zhengzhou, China; [2]Institute of Hepatobiliary and Pancreatic Diseases, Zhengzhou University, Zhengzhou, China; [3]Zhengzhou Basic and Clinical Key Laboratory of Hepatopancreatobiliary Diseases, Zhengzhou, China; [4]Department of Interventional Radiology, The First Affiliated Hospital of Zhengzhou University, Zhengzhou, China; [5]Department of Neurosurgery, The Fifth Affiliated Hospital of Zhengzhou University, Zhengzhou, China

*For correspondence:
liuzaoqu@163.com (ZL);
fcchanxw@zzu.edu.cn (XH);
ylsun@zzu.edu.cn (Y-lingS)

†These authors contributed equally to this work

Competing interest: The authors declare that no competing interests exist.

**Abstract** As the most aggressive tumor, the outcome of pancreatic cancer (PACA) has not improved observably over the last decade. Anatomy-based TNM staging does not exactly identify treatment-sensitive patients, and an ideal biomarker is urgently needed for precision medicine. Based on expression files of 1280 patients from 10 multicenter cohorts, we screened 32 consensus prognostic genes. Ten machine-learning algorithms were transformed into 76 combinations, of which we selected the optimal algorithm to construct an artificial intelligence-derived prognostic signature (AIDPS) according to the average C-index in the nine testing cohorts. The results of the training cohort, nine testing cohorts, Meta-Cohort, and three external validation cohorts (290 patients) consistently indicated that AIDPS could accurately predict the prognosis of PACA. After incorporating several vital clinicopathological features and 86 published signatures, AIDPS exhibited robust and dramatically superior predictive capability. Moreover, in other prevalent digestive system tumors, the nine-gene AIDPS could still accurately stratify the prognosis. Of note, our AIDPS had important clinical implications for PACA, and patients with low AIDPS owned a dismal prognosis, higher genomic alterations, and denser immune cell infiltrates as well as were more sensitive to immunotherapy. Meanwhile, the high AIDPS group possessed observably prolonged survival, and panobinostat may be a potential agent for patients with high AIDPS. Overall, our study provides an attractive tool to further guide the clinical management and individualized treatment of PACA.

## Editor's evaluation

This work sets out to develop a better machine learning-based predictor of survival/prognosis for patients diagnosed with pancreatic cancer. To achieve this, the authors developed a large combinatorial family of machine learning methods based on a high-dimensional set of -omics and other patient data features; using ten publicly available data sets. By finding the combined ML method(s) that performed best on the task, the authors were able to identify a reduced set of features (giving rise to a signature called AIDPS that involves 9 genes) which, when measured in the patient, allow

for fairly accurate prediction of patient survival and prognosis. Importantly, three new external data sets GSE21501, GSE57495, GSE71729 were used in the validation.

## Introduction

As the most aggressive tumor, pancreatic cancer (PACA) has a 5-year survival rate of only 11% and ranks fourth among tumor-related deaths in the United States (*Siegel et al., 2022*). Due to its insidious clinical manifestations and lack of available early detection and screening tools, 80–85% of PACA patients have progressed or metastasized at the time of detection, and lost the opportunity for surgical resection (*Mizrahi et al., 2020*). Over the past decade, immunotherapy, especially immune checkpoint inhibitors (ICIs), has made encouraging progress in most solid tumors (*Billan et al., 2020*). Unfortunately, ICIs have yielded disappointing clinical results in PACA because of the complex composition and highly suppressive immune microenvironment (*Bear et al., 2020*). In terms of molecularly targeted drugs, PARP inhibitors once shed light on *BRCA*-mutated PACA patients. However, a recent study confirmed that although the PARP inhibitor olaparib extended progression-free survival of patients (3.8 months vs. 7.4 months), the overall survival (OS) was not significantly improved (*Golan et al., 2019*). Reassuringly, the result of a phase Ib multicenter study showed that the CD40 monoclonal antibody APX005M in combination with chemotherapy achieved a 58% response rate in advanced PACA (*O'Hara et al., 2021*). Thus, in the era of precision medicine, it is very urgent to explore novel individualized management and combination therapy strategies to markedly improve the prognosis of PACA patients.

In clinical practice, the decision-making, therapeutic management, and follow-up still rely on the traditional anatomy-based TNM staging system (*Katz et al., 2008*). Although this provides a relatively trustworthy reference for determining whether patients will undergo surgical resection, the high inter- and intra-tumoral heterogeneity of PACA results in a wide range of outcomes even among patients at the same stage (*Liu et al., 2022c*). With the advancement of high-throughput sequencing and evidence-based medicine, molecular biomarkers such as *BRCA1/2* mutations, *NTRK* fusion, DNA mismatch repair deficiency (dMMR), and microsatellite instability-high (MSI-H) have been gradually brought into clinical guideline (*Wattenberg et al., 2020*; *Doebele et al., 2020*; *Le et al., 2017*). However, given the relatively low incidence but extremely high mortality of PACA, coupled with the current lack of optimal biomarkers to guide treatment decisions, patients may be over- or undertreated, resulting in heavy socioeconomic burden, serious toxic side effects, or rapid disease progression (*De Dosso et al., 2021*). In response to this problem, numerous multigene panels have been developed to address the wide heterogeneity of PACA and achieve relatively good performance in certain cohorts (*Wang et al., 2022*; *Tan et al., 2020*; *Yuan et al., 2021*). Considering that these prognostic models were based on the expression files of mRNAs, miRNAs, or lncRNAs in a specific pathway (e.g., immunity, metabolic reprogramming, m6A methylation), data utilization is insufficient (*Wang et al., 2022*; *Tan et al., 2020*; *Yuan et al., 2021*). In addition, due to uniqueness and inappropriateness of selected modeling methods, and the lack of strict validation in large multicenter cohorts, expression-based multigene signatures have great shortcomings thereby limiting their wide application in clinical settings (*Yokoyama et al., 2020*).

To develop an ideal biomarker, based on 32 consensus prognostic genes (CPGs), we constructed and multicenter validated a nine-gene artificial intelligence-derived prognostic signature (AIDPS) via 76 machine-learning algorithm combinations. In 13 multicenter cohorts, AIDPS exhibited robust performance in predicting OS, relapse-free survival (RFS), immunotherapy, and drug efficacy. After incorporating several vital clinicopathological features and 86 published signatures of PACA, our AIDPS also demonstrated stable and dramatically superior predictive capability. In addition, in other common digestive system tumors such as liver hepatocellular carcinoma (LIHC), stomach adenocarcinoma (STAD), colon adenocarcinoma (COAD), and rectum adenocarcinoma (READ), the AIDPS could still accurately stratify the prognosis. Overall, our study provides an important reference for achieving early diagnosis, prognostic evaluation, stratified management, individualized treatment, and follow-up of PACA in clinical practice.

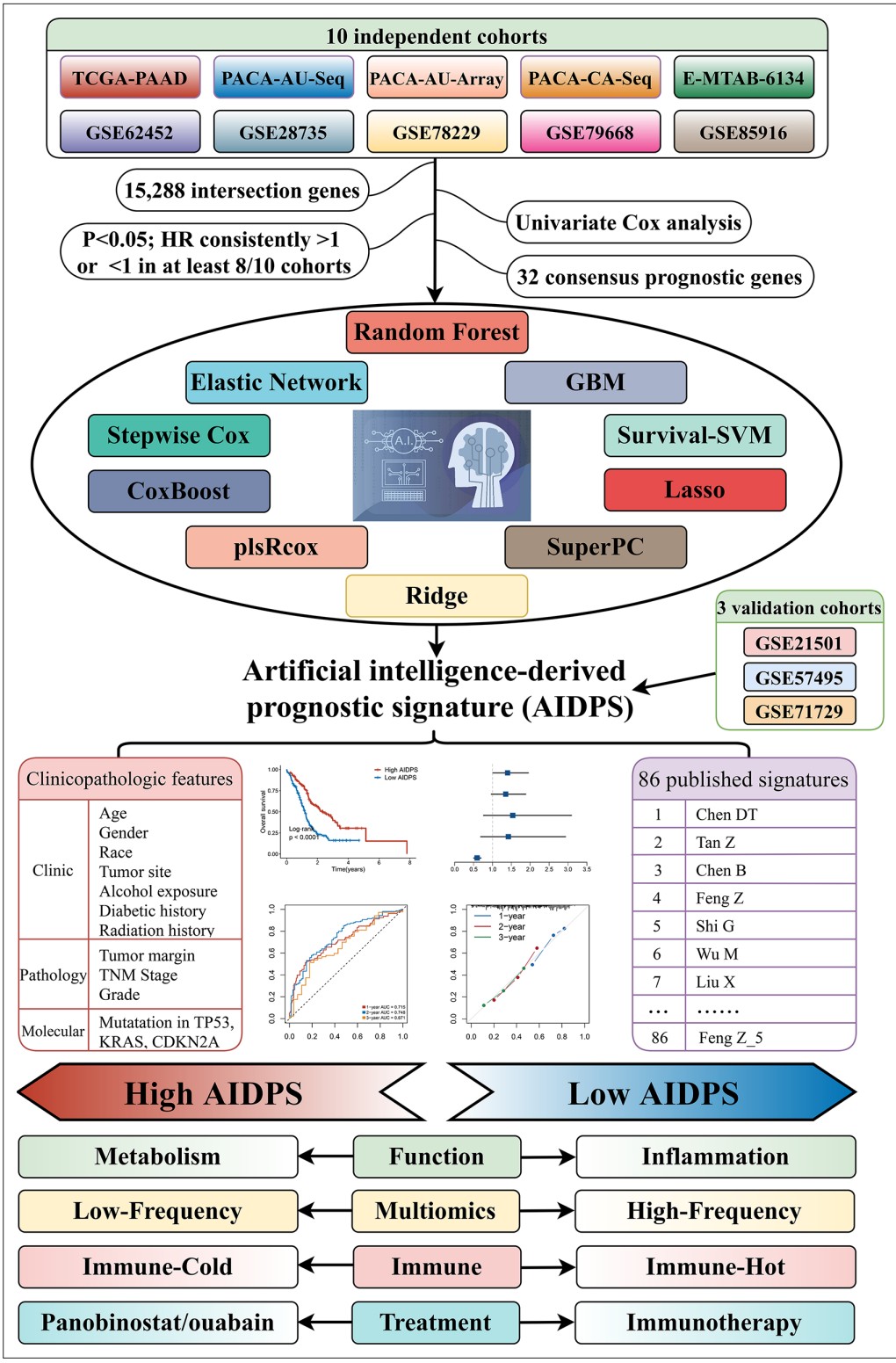

**Figure 1.** The workflow of our research.

The online version of this article includes the following source data for figure 1:

**Source data 1.** Details of baseline information in 13 public datasets.

# Results

## Integrated development of a pancreatic cancer consensus signature

Our workflow is outlined in *Figure 1*. Based on univariate Cox regression, we screened 32 CPGs from 15,288 intersection genes in the training and nine testing cohorts (*Figure 2B*). Next, these 32 CPGs were further incorporated into our integration program to develop an AIDPS. In the PACA-AU-Array training cohort, we applied 76 algorithm combinations via ten-fold cross-validation to construct prediction models and calculated the average C-index of each algorithm in the remaining nine testing cohorts. As shown in *Figure 2A*, the combination of CoxBoost and Survival-SVM with the highest average C-index (0.675) was selected as the final model. We further calculated AIDPS scores of each sample in all 13 cohorts according to the expression files of nine genes included in the AIDPS (*Figure 2—source data 1*).

## Consistent prognostic value of AIDPS

To evaluate the prognostic performance of AIDPS, we categorized PACA patients into high and low AIDPS groups according to the median value. The Kaplan–Meier curve for OS and RFS demonstrated the high AIDPS group possessed significantly longer survival in the PACA-AU-Array training cohort (p<0.0001 in OS and p=0.012 in RFS, *Figure 3A and B*). After removing batch effects, the Meta-Cohort combining 10 cohorts (training and nine testing cohorts) also exhibited the same trend (all p<0.05, *Figure 3C and D*). In addition, we further enrolled several important clinical traits for multivariate Cox analysis, and the results indicated that AIDPS was an independent protective factor for OS and RFS in the PACA-AU-Array cohort (HR: 0.593 [0.504–0.697] for OS and 0.762 [0.611–0.949] for RFS, both p<0.05, *Figure 3E and F*). Similar results were also found in the Meta-Cohort (HR: 0.603 [0.531–0.685] for OS and 0.667 [0.552–0.805] for RFS, both p<0.05, *Figure 3G and H*).

In the nine testing cohorts, Kaplan–Meier curves consistently showed a significantly prolonged OS in the high AIDPS group compared with the low AIDPS group (all p<0.05, *Figure 3—figure supplement 1A–I*). Similarly, the comparison of RFS also demonstrated that patients in the high AIDPS group possessed dramatically lower recurrence rate than low AIDPS group in the TCGA-PAAD (n = 69, p=0.029), PACA-CA-Seq (n = 113, p=0.0023), and E-MTAB-6134 (n = 288, p<0.0001) cohorts (*Figure 3—figure supplement 1J, L, M*). It is worth mentioning that only 28 samples in the PACA-AU-Seq owned complete RFS information. Although Kaplan–Meier analysis showed a corresponding trend, the log-rank test did not reach statistical significance (p=0.063, *Figure 3—figure supplement 1K*). After adjustment for available clinicopathological features, such as age, gender, TNM stage, grade, surgical margin, history of radiotherapy or alcohol consumption and *KRAS*, *TP53*, or *CDKN2A* mutations, multivariate Cox analysis results still indicated that AIDPS was an independent prognostic factor for OS (all p<0.05, *Figure 3—figure supplements 1N and 2A–F*). Consistently, the multivariate results of RFS also revealed that AIDPS remained statistically significant in the TCGA-PAAD, PACA-CA-Seq, and E-MTAB-6134 (all p<0.05, *Figure 3—figure supplement 2G–I*). However, given the small sample size of PACA-AU-Seq, the p-value was not statistically significant (p=0.338, *Figure 3—figure supplement 2J*).

## Robust predictive performance of AIDPS

To measure the discrimination of AIDPS, we plotted calibration curves and receiver-operator characteristic (ROC) curves. The calibration curves of both the PACA-AU-Array training cohort and Meta-Cohort showed that AIDPS had good prediction performance (*Figure 3I and J*). The area under the ROC curve (AUCs) of 1-, 2-, and 3-year OS were 0.715, 0.748, and 0.671 in the PACA-AU-Array training cohort and 0.717, 0.719, and 0.719 in the Meta-Cohort (*Figure 3K and L*). Similarly excellent results were found in the nine testing cohorts, with 0.705, 0.711, and 0.797 in the TCGA-PAAD; 0.749, 0.808, and 0.827 in the PACA-AU-Seq; 0.662, 0.683, and 0.691 in the PACA-CA-Seq; 0.773, 0.698, and 0.675 in the E-MTAB-6134; 0.676, 0.787, and 0.834 in the GSE62452; 0.734, 0.865, and 0.871 in the GSE28735; 0.669, 0.809, and 0.844 in the GSE78229; 0.791, 0.761, and 0.786 in the GSE79668; and 0.748, 0.766, and 0.811 in the GSE85916, respectively (*Figure 3—figure supplement 3A–I*). The results of AUCs greater than 0.65 in multiple independent cohorts indicated that our AIDPS could stably and robustly predict the prognosis of PACA patients.

In clinical settings, certain clinicopathological features such as surgical margin, stage, and grade are used for prognostic evaluation, clinical stratification management, and treatment decision-making

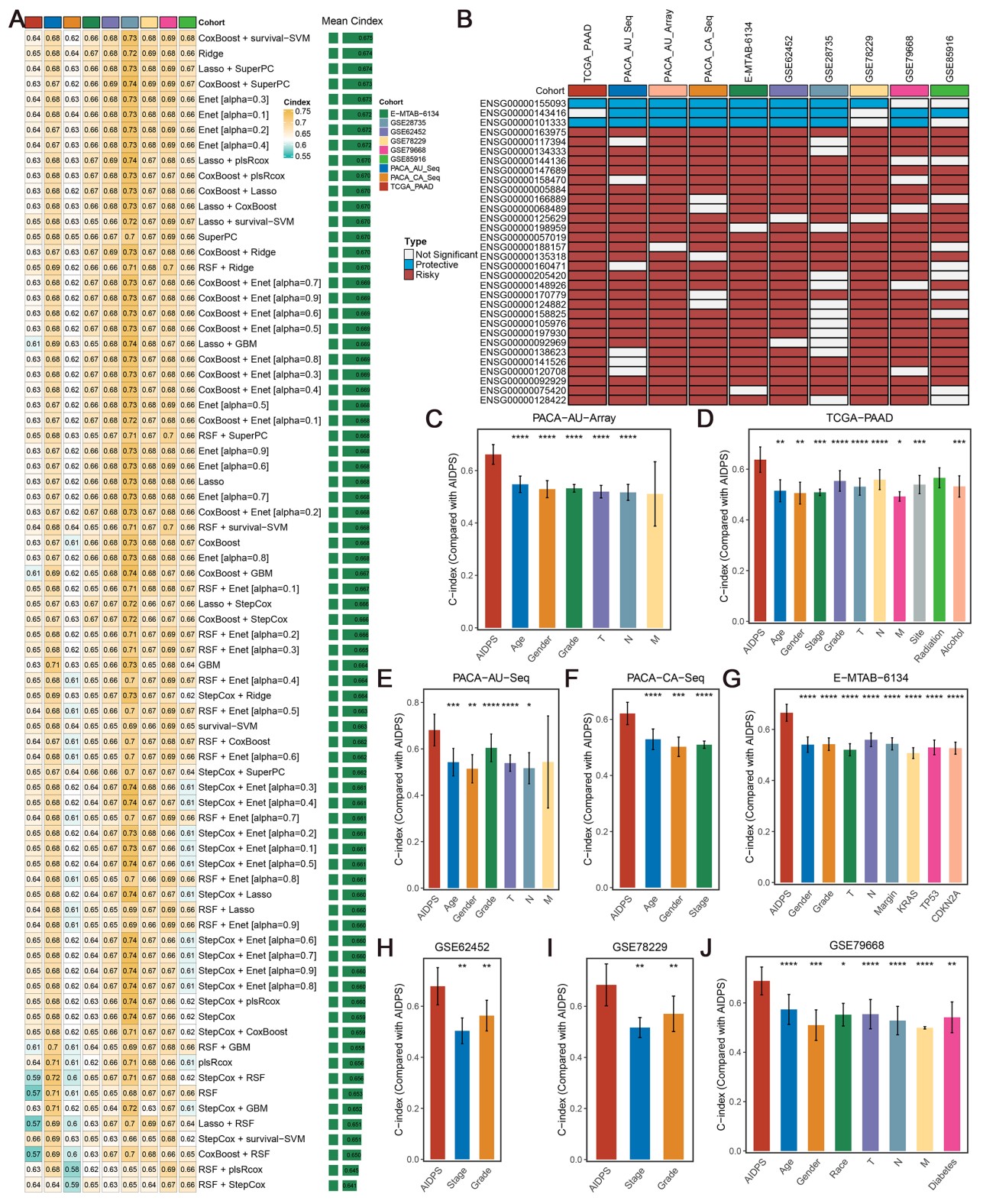

**Figure 2.** Construction and testing of the artificial intelligence-derived prognostic signature (AIDPS). (**A**) The C-indexes of 76 machine-learning algorithm combinations in the nine testing cohorts. (**B**) Discovery of 32 consensus prognosis genes from 10 independent multicenter cohorts. (**C–J**) The predictive performance of AIDPS was compared with common clinical and molecular variables in the PACA-AU-Array (**C**), TCGA-PAAD (**D**), PACA-AU-Seq (**E**), PACA-CA-Seq (**F**), E-MTAB-6134 (**G**), GSE62452 (**H**), GSE78229 (**I**), and GSE79668 (**J**). Z-score test: *p<0.05, **p<0.01, ***p<0.001, ****p<0.0001.

*Figure 2 continued on next page*

*Figure 2 continued*

The online version of this article includes the following source data for figure 2:

**Source data 1.** The nine genes included in the artificial intelligence-derived prognostic signature (AIDPS).

(*Ferrone et al., 2005*). Therefore, we contrasted the predicted efficacy of AIDPS and these common clinical traits in the eight cohorts containing clinical information. The results of C-index indicated that AIDPS had significantly improved accuracy than these features, including age, gender, race, diabetic history, TNM stage, grade, primary location, history of radiotherapy or alcohol consumption, surgical margin, and *KRAS*, *TP53* or *CDKN2A* mutations (*Figure 2C–J*).

In addition, to validate our model more rigorously, we evaluated the predictive performance of AIDPS in the validation cohorts. Kaplan–Meier survival analysis demonstrated that the high AIDPS group owned significantly prolonged OS in the three external validation cohorts (log-rank p=0.0014 in the GSE21501, 0.00045 in the GSE57495, 0.00011 in the GSE71729, *Figure 3—figure supplement 4B–D*). The AUCs of 1-, 2-, and 3-year OS were 0.677, 0.681, and 0.761 in the GSE21501, 0.682, 0.728, and 0.747 in the GSE57495; 0.676, 0.693, and 0.714 in the GSE71729 (*Figure 3—figure supplement 4E–G*). The calibration curves also confirmed the good predictive performance of AIDPS (*Figure 3—figure supplement 4H–J*).

Overall, Kaplan–Meier survival analysis, Cox regression analysis, timeROC curve, C-index, and calibration curve results of the training cohort, nine testing cohorts, Meta-Cohort, and three external validation cohorts consistently indicated that AIDPS could accurately and robustly predict the prognosis of PACA patients, suggesting that AIDPS may become an attractive tool to serve clinical practice.

## Re-evaluation of previously 86 published signatures in PACA

The rapid development of high-throughput sequencing has shed light on the stratified management and precise treatment of tumors. In recent years, numerous prognostic signatures of PACA have been constructed via machine-learning algorithms such as LASSO and Stepwise Cox based on large amounts of high-quality data (*Wang et al., 2022*; *Tan et al., 2020*; *Yuan et al., 2021*). Therefore, we additionally collected 86 published mRNA/lncRNA prognostic signatures to compare the predictive accuracy of AIDPS and these models (*Figure 4—source data 1*). Signatures containing miRNAs were excluded due to the lack of necessary miRNA expression information. The results of univariate Cox regression showed that only our AIDPS and 20-gene signature of Demirkol CS had consistent statistical significance in all 13 independent cohorts and Meta-Cohort (*Figure 4A*, *Figure 3—figure supplement 4A*).

We then compared the predictive power of AIDPS and these 86 signatures via C-index across the training cohort, nine testing cohorts, and Meta-Cohort (*Figure 4B*). Our AIDPS exhibited distinctly superior accuracy than the other models in almost all cohorts (ranked first in four cohorts, ranked second in three cohorts, and ranked third in two cohorts), revealing the robustness of AIDPS. Of note, various prognostic signatures owned higher C-index in their TCGA-PAAD training cohort (e.g., Zhang C, Xu Q, Li Z, etc.), but performed poorly in other cohorts, which might be due to impaired generalization ability from overfitting (*Figure 4B*).

In addition, our AIDPS also possessed robust predictive performance across the three external validation cohorts, ranking fourth in the GSE57495 cohort as well as fifth in the GSE21501 and GSE71729 cohorts, which outperformed almost all published signatures (*Figure 4—figure supplement 1A–C*). Notably, although the six-gene signature of Stratford JK was significantly better than AIDPS in the GSE21501 and GSE71729 cohorts, it was constructed in the GSE21501 cohort and performed very poorly in other cohorts, with C-index even less than 0.6 in the GSE57495, TCGA-PAAD, PACA-AU-Seq, etc. (*Figure 4—figure supplement 1E*). The 15-gene signature of Chen DT owned observably superior performance in his own training cohort GSE57495, but it was unsatisfactory in the GSE21501, GSE71729, PACA-CA-Seq, and other cohorts (*Figure 4—figure supplement 1F*). Similarly, Kim J's five-gene signature performed well in the training cohort GSE71729 as well as GSE21501 and PACA-AU-Seq cohorts, but very poorly in most other cohorts such as GSE85916, GSE57495, PACA-CA-Seq, and E-MTAB-6134 (*Figure 4—figure supplement 1G*). Furthermore, although the 20-gene signature of Demirkol CS, the 3-gene signature of Chen H, the 6-gene signature of Hou J, the 7-gene signature of Liu X, and the 6-gene signature of Yu X_2 were superior to AIDPS in their training cohort or a few other cohorts, only our AIDPS possessed acceptable performance in all PACA cohorts, and the vast

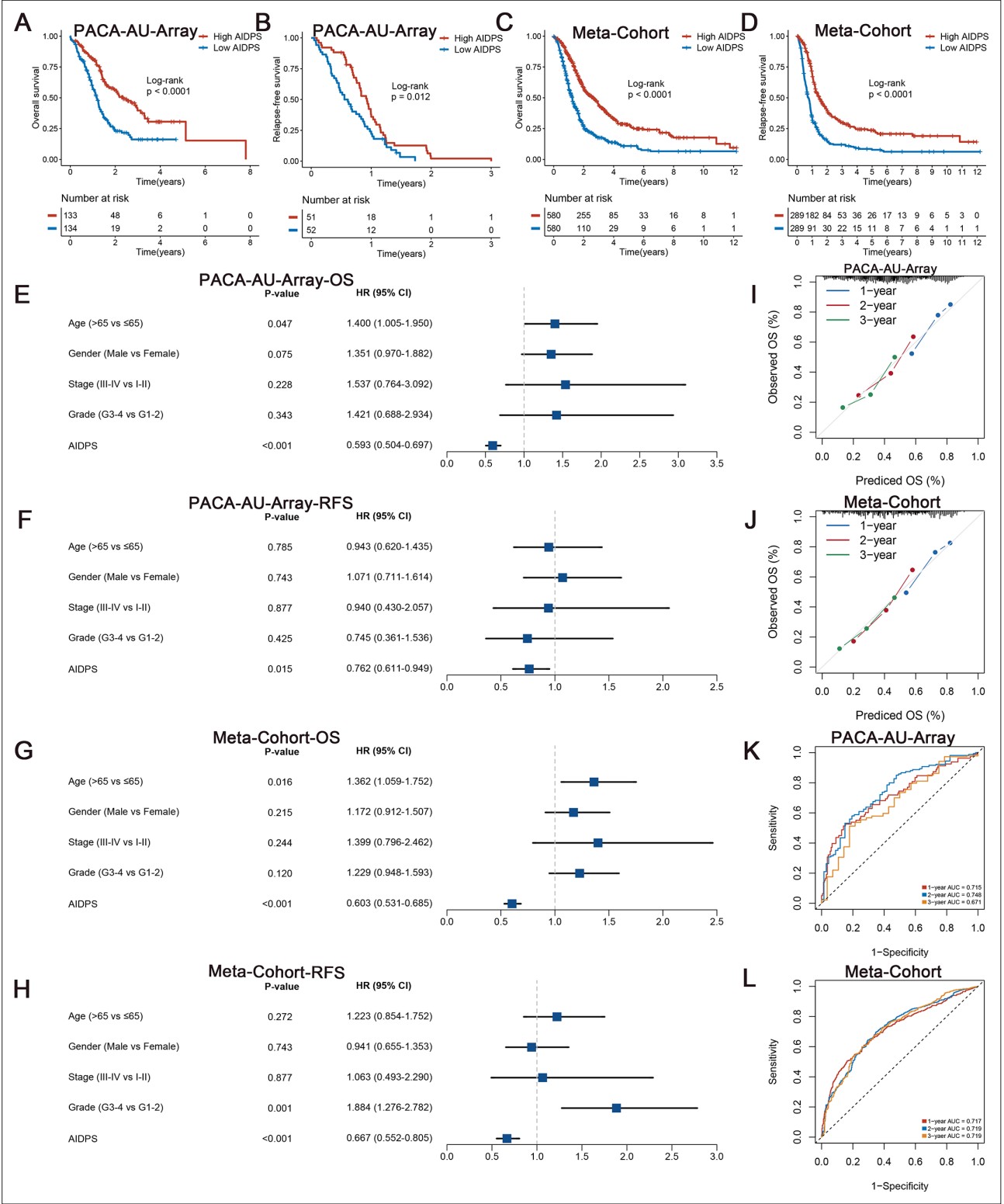

**Figure 3.** Survival analysis and predictive performance evaluation of artificial intelligence-derived prognostic signature (AIDPS). (**A, B**) Kaplan–Meier survival analysis for overall survival (OS) (**A**) and relapse-free survival (RFS) (**B**) between the high and low AIDPS groups in the PACA-AU-Array. (**C, D**) Kaplan–Meier survival analysis for OS (**C**) and RFS (**D**) between the high and low AIDPS groups in the Meta-Cohort. (**E, F**) Multivariate Cox regression analysis of OS (**E**) and RFS (**F**) in the PACA-AU-Array. (**G, H**) Multivariate Cox regression analysis of OS (**G**) and RFS (**H**) in the Meta-Cohort. (**I, J**) Calibration curve for predicting 1-, 2-, and 3-year OS in the PACA-AU-Array (**I**), and Meta-Cohort (**J**). (**K, L**) Time-dependent receiver-operator characteristic (ROC) analysis for predicting 1-, 2-, and 3-year OS in the PACA-AU-Array (**K**), and Meta-Cohort (**L**).

*Figure 3 continued on next page*

*Figure 3 continued*

The online version of this article includes the following figure supplement(s) for figure 3:

**Figure supplement 1.** Survival analysis of artificial intelligence-derived prognostic signature (AIDPS) in the nine testing cohorts.

**Figure supplement 2.** Survival analysis of artificial intelligence-derived prognostic signature (AIDPS) in the nine testing cohorts.

**Figure supplement 3.** Predictive performance of artificial intelligence-derived prognostic signature (AIDPS) in the nine testing cohorts.

**Figure supplement 4.** Survival analysis and predictive performance of artificial intelligence-derived prognostic signature (AIDPS) in the three external validation cohorts.

majority of cohorts had good performance with a C-index greater than 0.65 (*Figure 4B*, *Figure 4— figure supplement 1A–C*). In conclusion, the above results suggested that the nine-gene AIDPS could robustly predict the prognosis of PACA patients, and fewer genes might make it more valuable for clinical promotion.

## The clinical signature of AIDPS

We further compared several familiar clinical characteristics between the high and low AIDPS groups, and the results indicated the absence of statistical differences in age, gender, and TNM stage (*Figure 5A–C*, *Figure 5—figure supplement 1A–L*). However, patients with low AIDPS possessed more advanced grades, which might contribute to their worse prognosis (*Figure 5D*, *Figure 5— figure supplement 1M–P*).

In addition, given the excellent predictive power of AIDPS in PACA, we additionally tested its performance in several other common digestive system tumors. As shown in *Figure 5E–H*, the Kaplan–Meier survival curves exhibited significantly dismal OS for patients in the low AIDPS group in four tumors, including LIHC (p=0.016), STAD (p=0.037), COAD (p=0.032), and READ (p=0.026). These results supported our hypothesis, suggesting that AIDPS constructed in PACA, as a biomarker, has broad prospects for generalization to other tumors.

## The underlying biological mechanisms of AIDPS

Gene set enrichment analysis (GSEA) was applied to elucidate the potential functional pathways of AIDPS. As illustrated in *Figure 5I and J*, the high AIDPS group was remarkably enriched for digestive and metabolism-related pathways, such as insulin secretion and regulation, peptide hormone secretion and regulation, fat digestion and absorption, pancreatic secretion, maturity onset diabetes of the young, and type Ⅱ diabetes mellitus. While the low AIDPS group was predominantly correlated with the regulation of T cell proliferation, *IL-17* signaling pathway, and other immune-related pathways, as well as cell cycle, nuclear chromosome segregation, homologous recombination, and other proliferation-related biological processes, which partly explained its more advanced grades and worse prognosis (*Figure 5K and L*).

## Genome alteration landscape of AIDPS

To investigate genomic heterogeneity between the high and low AIDPS groups, we performed a comprehensive analysis of mutations and copy number alteration (CNA, *Figure 6A*). As shown in *Figure 6C*, the low AIDPS group possessed observably higher tumor mutation burden (TMB). Combining the 10 oncogenic signaling pathways in TCGA (*Sanchez-Vega et al., 2018*), we found that the classical tumor suppressor genes *TP53*, *CDKN2A*, and oncogene *KRAS* were more frequently mutated in the low AIDPS group than high AIDPS group, whereas the opposite was true for *SMAD4*, *TTN*, and *RNF43* (*Figure 6A and B*). Next, based on the popular mutational signatures in PACA, we discovered that mutational signature 3 (*BRCA1/2* mutations-related) was enriched in the high AIDPS group, while mutational signature 1 (age-related) was more dominant in low AIDPS group.

In addition, we further explored the CNA landscape of the two groups. Compared to the high AIDPS group, the low AIDPS group owned evidently higher amplification or deletion in the focal and chromosome arm levels, such as the amplification of 8q24.21, 19q13.2, and 8p11.22 as well as deletion of 9p21.3, 18q21.2, 6p22.2, and 22q13.31 (*Figure 6A and D*). This result was again corroborated in gene level by the obvious amplification of the oncogene *MYC* within 8q24.21, and the distinct deletion of the tumor suppressor genes *CDKN2A*, *CDKN2B*, and *SMAD4* within 9p21.3 and 18q21.2

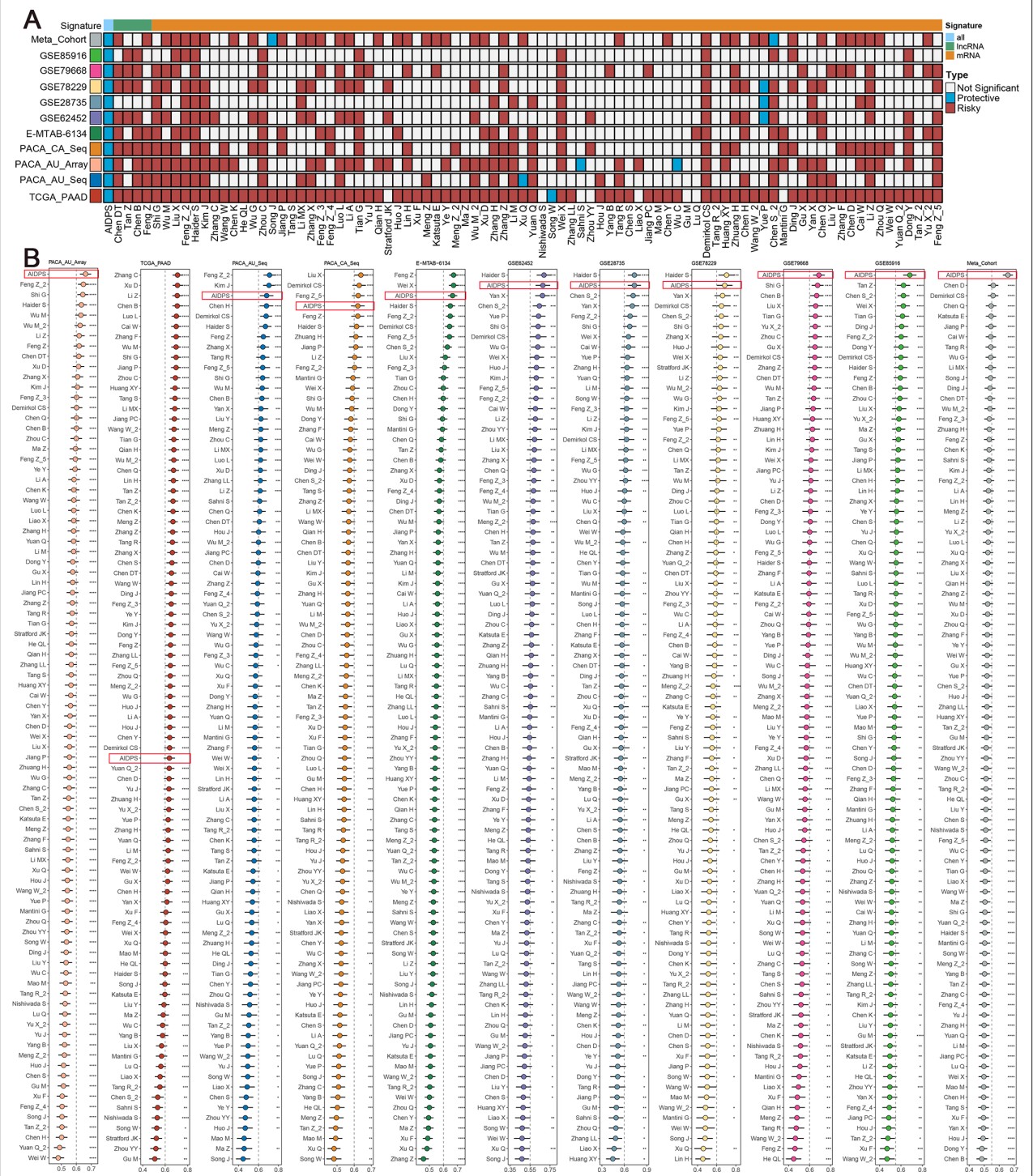

**Figure 4.** Comparisons between artificial intelligence-derived prognostic signature (AIDPS) and 86 expression-based signatures. (**A**) Univariate Cox regression analysis of AIDPS and 86 published signatures of pancreatic cancer (PACA). (**B**) C-indexes of AIDPS and 86 published signatures in the PACA-AU-Array, TCGA-PAAD, PACA-AU-Seq, PACA-CA-Seq, E-MTAB-6134, GSE62452, GSE28735, GSE78229, GSE79668, GSE85916, and Meta-Cohort. Z-score test: *p<0.05, **p<0.01, ***p<0.001, ****p<0.0001.

The online version of this article includes the following source data and figure supplement(s) for figure 4:

**Source data 1.** Details of 86 published mRNA/LncRNA signatures in pancreatic cancer (PACA).

**Figure supplement 1.** Comparison of artificial intelligence-derived prognostic signature (AIDPS) with 86 published signatures in the three validation cohorts and with models constructed by other methods for nine AIDPS genes.

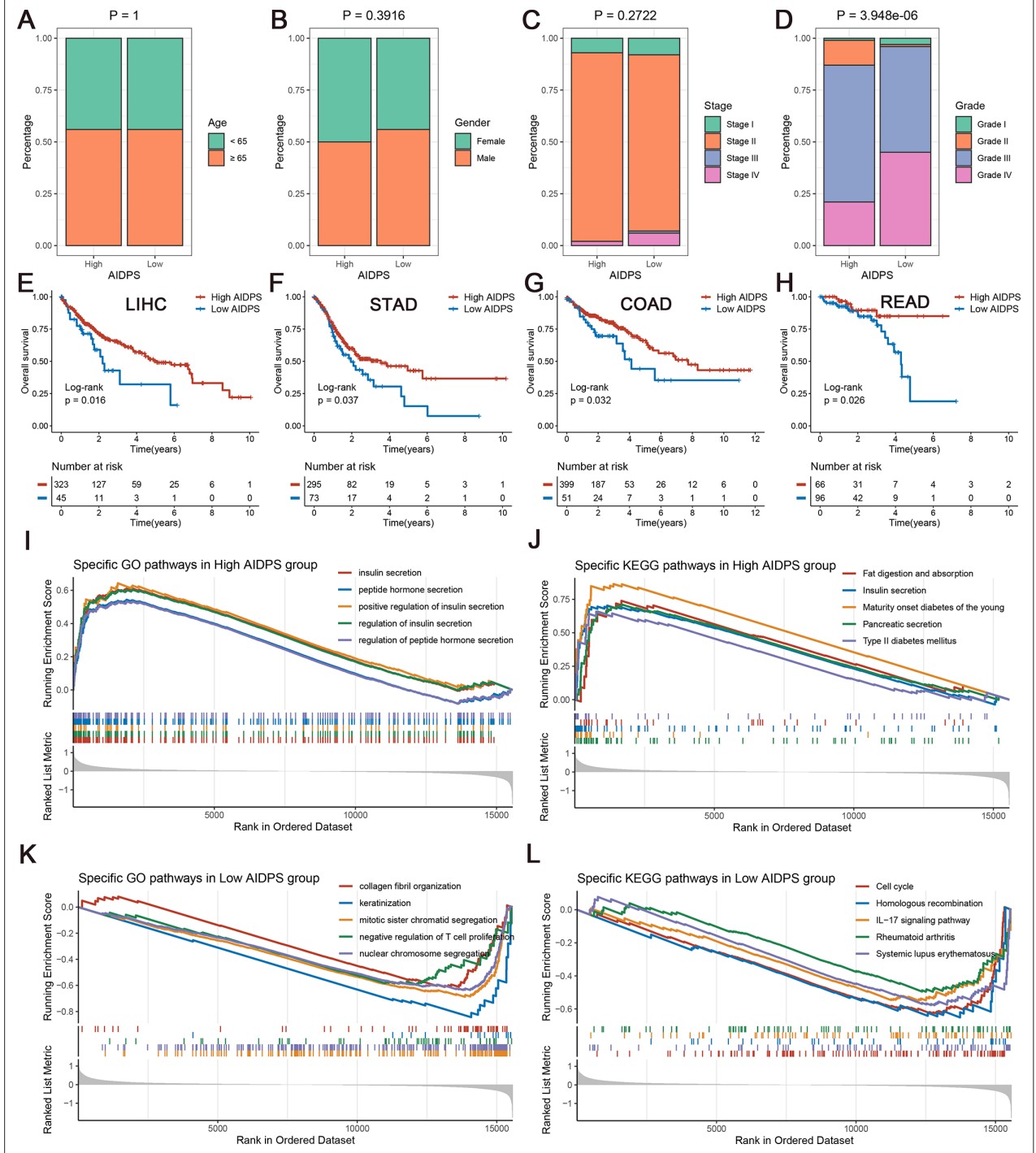

**Figure 5.** The clinical signature and functional characteristics of the high and low artificial intelligence-derived prognostic signature (AIDPS) groups. (A–D) Composition percentage of the two groups in clinical characteristics such as age (A), gender (B), stage (C), and grade (D) in the PACA-AU-Array. (E–H) Kaplan–Meier survival analysis for overall survival (OS) in the TCGA-LIHC (E), TCGA-STAD (F), TCGA-COAD (G), and TCGA-READ (H). (I, J) The top five Gene Ontology (GO)-enriched pathways (I) and Kyoto Encyclopedia of Genes and Genomes (KEGG)-enriched pathways (J) in the high AIDPS groups. (K, L) The top five GO-enriched pathways (K) and KEGG-enriched pathways (L) in the low AIDPS groups.

The online version of this article includes the following figure supplement(s) for figure 5:

**Figure supplement 1.** The clinical characteristics of the high and low artificial intelligence-derived prognostic signature (AIDPS) groups.

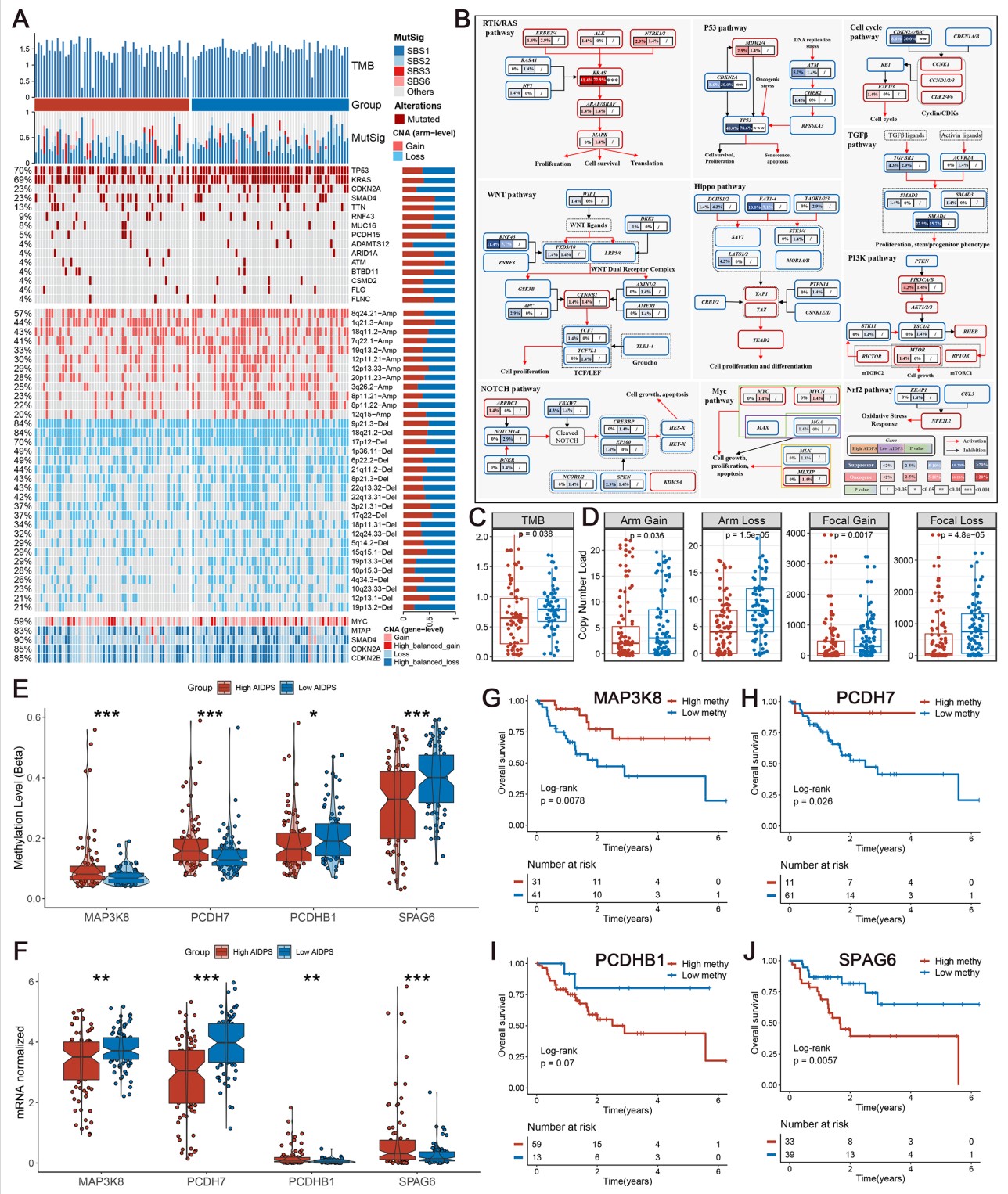

**Figure 6.** Multi-omics analysis based on mutation, copy number alteration (CNA), and methylation. (**A**) Genomic alteration landscape according to artificial intelligence-derived prognostic signature (AIDPS). Tumor mutation burden (TMB), relative contribution of four mutational signatures, top 15 mutated genes and broad-level CNA (>20%), and selected genes located within chromosomes 8q24.21, 9p21.3, and 18q21.2 are shown from the top to the bottom panels. The proportion of the high and low AIDPS groups in each alteration is presented in the right bar charts. (**B**) Comprehensive comparison of mutation landscapes in 10 oncogenic signaling pathways across the high and low AIDPS groups. Genes are mutated at different frequencies (color intensity indicates the mutation frequency within the entire dataset) by oncogenic mutations (red) and tumor suppressor mutations

*Figure 6 continued on next page*

Figure 6 continued

(blue). Each gene box includes two percentages representing the mutation frequency in the high and low AIDPS groups, and another box representing the statistical p-value. Genes are grouped by signaling pathways, with edges showing pairwise molecular interactions. (C) Comparison of the two groups in TMB. (D) Comparison of the two groups in arm and focal CNA burden. (E, F) Boxplot of DNA methylation level (E) and mRNA expression level (F) for methylation-driven genes in the high and low groups. (G–J) Kaplan–Meier survival analysis between the high and low methylation groups in the *MAP3K8* (G), *PCDH7* (H), *PCDHB1* (I), and *SPAG6* (J). *p<0.05, **p<0.01, ***p<0.001.

(*Figure 6A*). Overall, oncogenes amplification and tumor suppressor genes deletion in the low AIDPS group might contribute to their poor prognosis.

## Methylation-driven events of AIDPS

Referring to our previous process (*Liu et al., 2021b*; *Liu et al., 2021c*), we screened four methylation driver genes (MDGs) whose methylation levels were significantly inversely related to matched gene expression levels in PACA. Compared to the low AIDPS group, the high AIDPS group possessed higher *MAP3K8* and *PCDH7* methylation levels as well as significantly lower mRNA expression levels, while the opposite was true for *PCDHB1* and *SPAG6* (*Figure 6E and F*). Furthermore, the Kaplan–Meier analysis showed that higher methylation levels of *MAP3K8* and *PCDH7* and lower methylation level of *SPAG6* brought significantly prolonged OS for the high AIDPS group (all p<0.05, *Figure 6G, H and J*). *PCDHB1* also exhibited a similar trend with *SPAG6*, although statistical significance was not reached (p=0.07, *Figure 6I*).

## Immune landscape and molecular expression of AIDPS

The above GSEA revealed that several immune-related pathways were highly enriched in the low AIDPS group, and we consequently investigated the immune landscape and immune checkpoint molecules (ICMs) expression between the two groups. According to single-sample gene set enrichment analysis (ssGSEA), we found that the low AIDPS group exhibited a relatively higher infiltration abundance of immune cell types, including activated CD4+ T cells, CD56dim natural killer cells, central memory CD8+ T cells, gamma delta T cells, and type 2T helper cells (all p<0.05, *Figure 7A and B*). In addition, among the 27 ICMs we included, the low AIDPS group had dramatically increased relative expression level, such as *CD274*, *CD276*, *PDCD1LG2*, *CD40*, *CD70*, *TNFRSF18*, *TNFRSF4*, *TNFRSF9*, and *NT5E* (*Figure 7C*). Together, these results consistently indicated that PACA patients with low AIDPS were more likely to respond to immunotherapy.

## Prognostic value and biological relevance of nine AIDPS genes

The currently known biological functions of these nine AIDPS genes are summarized in Appendix 1. Based on large multicenter survival data from the training cohort, nine testing cohorts, and three external validation cohorts, we performed an integrated univariate Cox regression analysis of survival variables using AIDPS and its nine genes as continuous variables. As shown in *Figure 7—figure supplement 1A*, AIDPS was an independent protective factor in all 13 cohorts (consistent with *Figure 3—figure supplement 1K*, only 28 samples in the PACA-AU-Seq with RFS information, and although exhibited a corresponding trend, it did not reach statistical significance). Correspondingly, because the 32 CPGs used to construct AIDPS with consistent prognostic value in at least 8/10 cohorts, nine AIDPS genes had relatively stable performance in the training and nine testing cohorts (*Figure 7—figure supplement 1B–J*). However, in the three external validation cohorts, the poor performance of these nine AIDPS genes was hardly satisfactory. Furthermore, based on the expression files of these nine AIDPS genes in the PACA-AU-Array training cohort, we constructed 18 models via 10 common machine-learning algorithms (α values for elastic networks from 0.1 to 0.9). As expected, among all 18 models, the model built by survival-SVM, namely, our nine-gene AIDPS, achieved a maximum mean C-index of 0.666 in the remaining 12 multicenter cohorts (*Figure 4—figure supplement 1D*). That is, for the nine genes obtained from 32 CPGs after dimensionality reduction by CoxBoost, these results reconfirmed our previous pipeline results (*Figure 2A*), and the AIDPS constructed by survival-SVM was the best choice. Overall, our AIDPS brings significant performance improvements compared to the nine AIDPS genes alone.

To describe the biological relevance between AIDPS and its nine genes, we performed Pearson correlation analysis referring to previous studies (*Zhang et al., 2020*). The results showed that

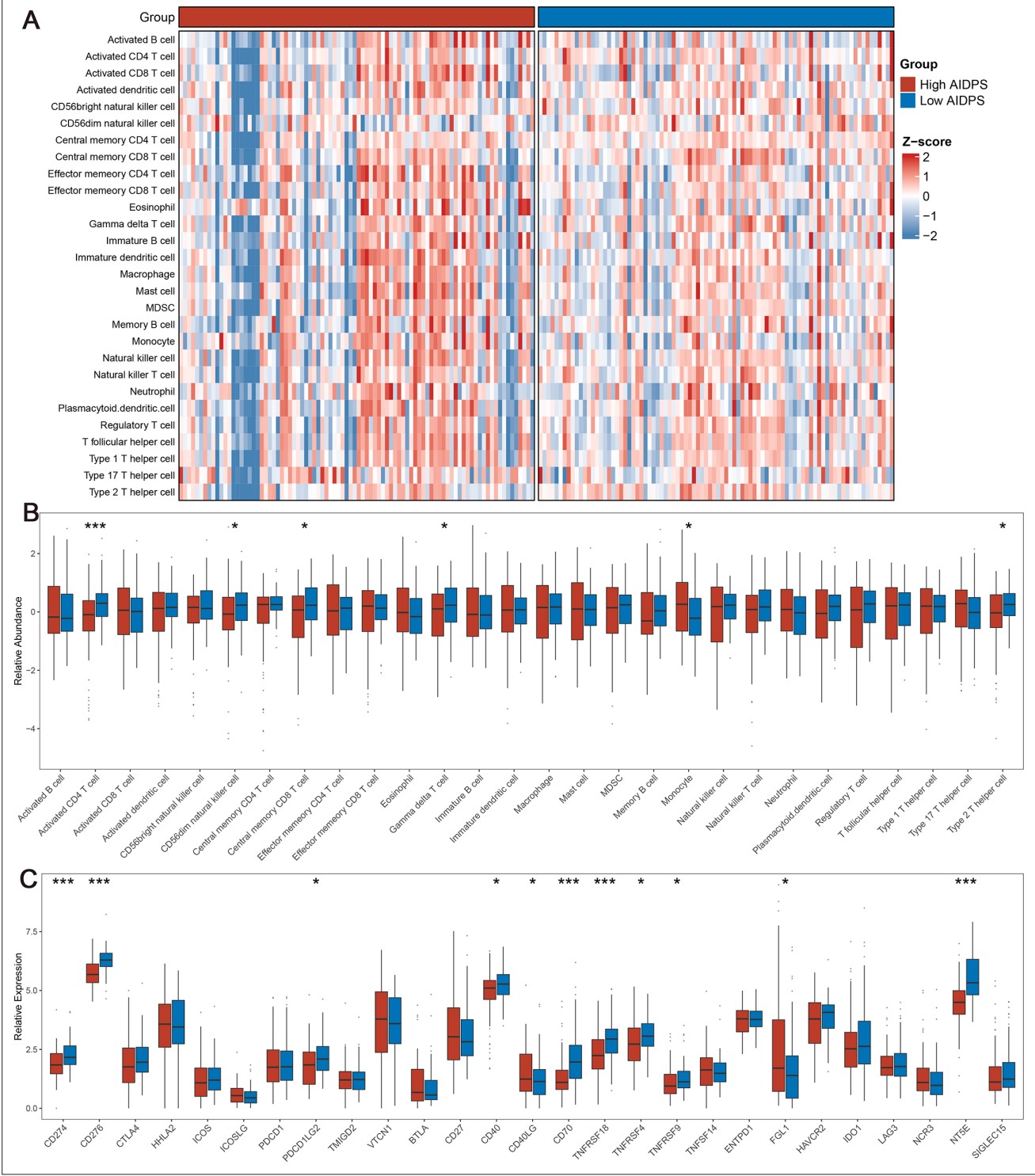

**Figure 7.** The immune landscape in the high and low artificial intelligence-derived prognostic signature (AIDPS) groups. (**A**) The heatmap of 28 immune cell types in the high and low AIDPS groups. (**B**) Boxplot of relative infiltrate abundance of 28 immune cell types in patients with high and low AIDPS groups. (**C**) Boxplot of relative expression levels at 27 immune checkpoints profiles between the high and low AIDPS patients. *p<0.05, **p<0.01, ***p<0.001.

The online version of this article includes the following figure supplement(s) for figure 7:

**Figure supplement 1.** Integrated Cox regression analysis of survival variables for artificial intelligence-derived prognostic signature (AIDPS) and nine AIDPS genes in 13 cohorts.

*Figure 7 continued on next page*

*Figure 7 continued*

**Figure supplement 2.** The correlation of artificial intelligence-derived prognostic signature (AIDPS) and its nine genes with immune molecules and immune cell types.

**Figure supplement 3.** The relevance of artificial intelligence-derived prognostic signature (AIDPS) and its nine genes with mutation and copy number alteration.

*SELENBP1* and *PLCB4*, which were significantly positively correlated with AIDPS, had a moderate negative correlation with seven genes that were significantly negatively correlated with AIDPS in the whole and low AIDPS TCGA-PAAD (*Figure 7—figure supplement 2A and C*). Interestingly, the overall correlation was lower in the high AIDPS group and numerous genes showed different or even opposite trends (e.g., *SELENBP1* changed from positive to negative correlation with AIDPS and *PLCB4*, *Figure 7—figure supplement 2B*). In addition, given the obvious impact of AIDPS in the tumor microenvironment (TME), we compared the relationship of AIDPS and its nine genes with 27 ICMs and 28 immune cell types in the whole, high AIDPS, and low AIDPS TCGA-PAAD. The Pearson correlation analysis exhibited that AIDPS and its positively correlated *SELENBP1* and *PLCB4* had a strong negative correlation with ICMs and immune cell types, while *DCBLD2, PRR11, UNC13D, EREG,* and *TGM2*, which were negatively correlated with AIDPS, were significantly positively correlated with ICMs and immune cell types in the whole and high AIDPS TCGA-PAAD (*Figure 7—figure supplement 2D, E, G, H*). However, the trend in the low AIDPS group was contrary, AIDPS and *PLCB4* were observably positively correlated with ICMs and immune cell types, while *DCBLD2, PRR11, UNC13D, EREG, ADM, CDCA4,* and *TGM2* were negatively correlated with ICMs and immune cell types (*Figure 7—figure supplement 2F and I*).

In terms of genomic alteration, we also observed that *SELENBP1* and *PLCB4*, which were positively correlated with AIDPS, were significantly lower in the low AIDPS group with superior mutation and CNA frequency, suggesting that they were significantly negatively correlated with TMB and CNA burden (*Figure 7—figure supplement 3*). Correspondingly, *DCBLD2, PRR11, UNC13D, EREG, ADM, CDCA4,* and *TGM2*, which were negatively correlated with AIDPS, were remarkably increased in the low AIDPS group, hinting they were significantly positively correlated with TMB and CNA burden (e.g., the low AIDPS group possessed higher *TP53* mutation and 8q24.21 amplification, as well as higher *DCBLD2* expression, *Figure 7—figure supplement 3*).

## Predictive value of AIDPS for immunotherapy

Given that patients in the low AIDPS group possessed higher genomic alteration frequency and TMB, combined with their relatively activated TME and increased ICMs expression, we speculated that PACA patients with low AIDPS were more sensitive to immunotherapy. Based on the Tumour Immune Dysfunction and Exclusion (TIDE) web tool, the low AIDPS group resulted in significantly lower TIDE scores and higher immunotherapy response rates (*Figure 8A and B*). The results of the Subclass Mapping (Submap) also suggested that expression patterns of patients with low AIDPS was more similar to those of melanoma patients who responded to ICIs (*Figure 8C*). Overall, these results demonstrated that the low AIDPS group was more likely to benefit from immunotherapy.

## Searching for potential therapeutic agents for the high AIDPS group

As illustrated in *Figure 8E*, we developed potential agents for PACA patients with high AIDPS using sensitivity data from Cancer Therapeutics Response Portal (CTRP, includes 481 compounds over 835 cancer cell lines [CCLs]) and profiling relative inhibition simultaneously in mixtures (PRISM) (includes 1448 compounds over 482 CCLs) datasets (*Yang et al., 2021*). To ensure the reliability of our protocol, gemcitabine, as a first-line treatment for PACA, was employed to investigate whether the estimated sensitivity and clinical practice were concordant. A laboratory study found that elevated *PAK1* activity was required for gemcitabine resistance in PACA, and that *PAK1* inhibition enhanced the efficacy of gemcitabine. Consistent with this study, our results revealed that patients with low *PAK1* expression possessed distinctly lower estimated AUC values, suggesting greater sensitivity to gemcitabine (*Figure 8F*). Afterward, we applied this formula to identify potentially sensitive agents for the high AIDPS group, and finally generated four CTRP-derived agents (brefeldin A, oligomycin A, ouabain, and panobinostat) and nine PRISM-derived agents (aspirin, BAY-87–2243, EVP4593, GSK2656157, I-BET151, LY303511, OTX015, oxaliplatin, and XL388). The estimated AUC values of these agents

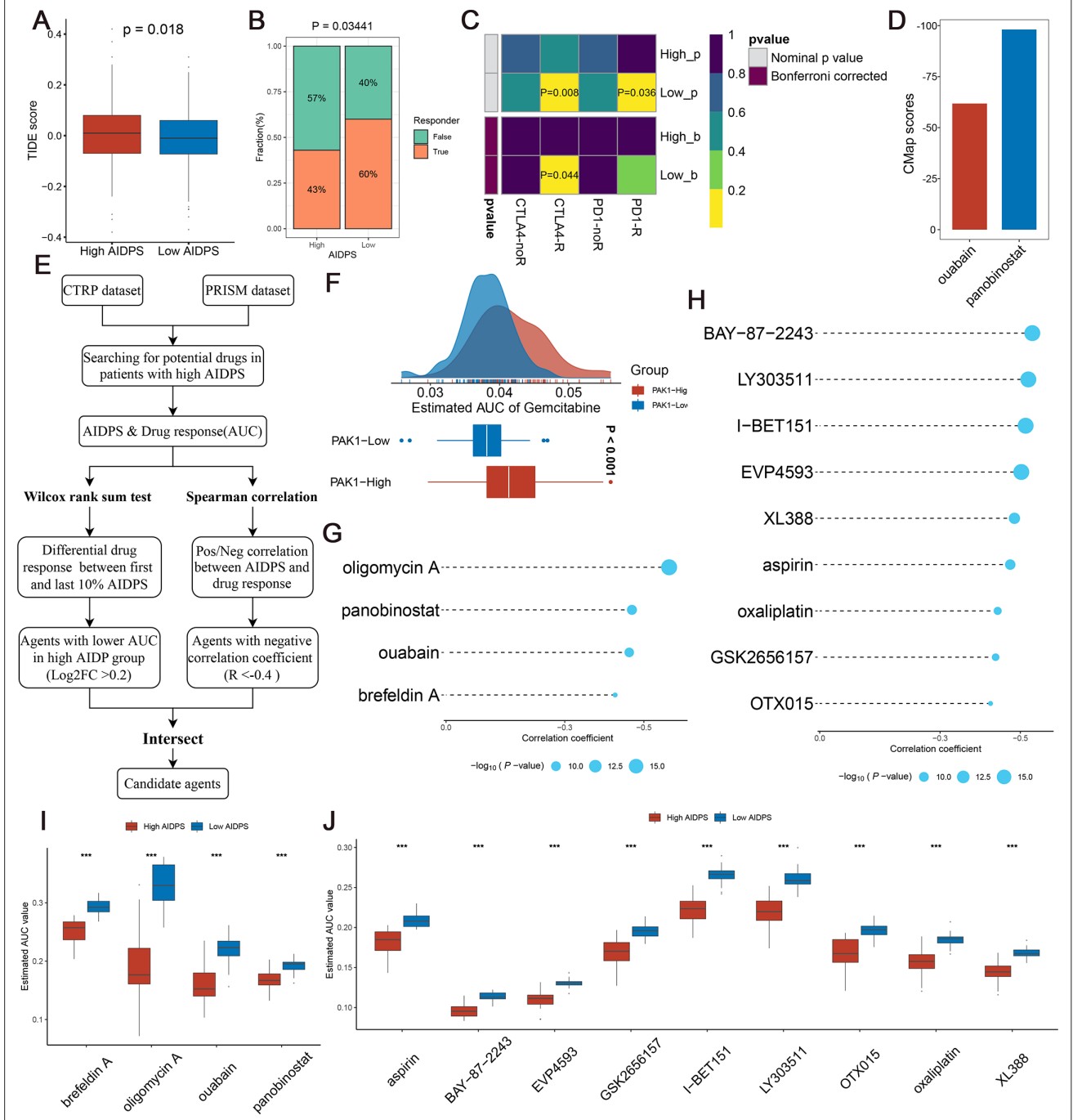

**Figure 8.** Evaluating therapeutic drug benefits. (**A**) Boxplot of Tumour Immune Dysfunction and Exclusion (TIDE) score between the high and low artificial intelligence-derived prognostic signature (AIDPS) groups. (**B**) Percentage of immunotherapy responses at high and low AIDPS groups. (**C**) Submap analysis of the two groups and 47 pretreated patients with comprehensive immunotherapy annotations. For Submap analysis, a smaller p-value implied a more similarity of paired expression profiles. (**D**) Barplot of ouabain and panobinostat CMap scores in patients with high AIDPS. (**E**) Schematic outlining the strategy to develop potential therapeutic agents with higher drug sensitivity in the high AIDPS group. (**F**) Comparison of estimated gemcitabine's sensitivity between high and low *PAK1* expression groups. (**G, H**) The results of Spearman's correlation analysis of Cancer Therapeutics Response Portal (CTRP)-derived compounds (**G**) and profiling relative inhibition simultaneously in mixtures (PRISM)-derived compounds (**H**). (**I, J**) The results of differential drug response analysis of CTRP-derived compounds (**I**) and PRISM-derived compounds (**J**), the lower values on the y-axis of boxplots imply greater drug sensitivity. CMap, Connectivity Map *p<0.05, **p<0.01, ***p<0.001.

were not only statistically negatively correlated with AIDPS scores, but also significantly lower in the high AIDPS group (*Figure 8G–J*).

In addition, based on the differential expression profiles between PACA patients and normal controls, we further applied the Connectivity Map (CMap, https://clue.io/) tool to identify candidate compounds for PACA. After taking the intersection with the results obtained by CTRP and PRISM, we ended up with two candidate compounds: ATPase inhibitor ouabain and histone deacetylase (HDAC) inhibitor panobinostat. Among them, panobinostat with a –98.11 CMap score was highly sensitive in PACA patients, suggesting that it could become a potential therapeutic agent for PACA patients in the high AIDPS group (*Figure 8D*).

## Discussion

Over the past 20 years, the incidence of PACA has increased by 0.5–1.0% per year, but the 5-year survival rate only improved from 5.26% to 10%, without a significant breakthrough (*Park et al., 2021*). The lack of available biomarkers for screening, stratified management, and prognostic follow-up has been an urgent problem for clinicians and researchers, which may lead to over- or undertreatment. To bridge these gaps, we constructed and validated a nine-gene AIDPS via 76 machine-learning algorithm combinations in 13 independent multicenter cohorts. Compared to several common clinico-pathological features and 86 published signatures of PACA, our AIDPS demonstrated robust and superior predictive capacity. Moreover, we substantiated that the low AIDPS group had a dismal prognosis, more advanced grades, enrichment in immune and proliferation-related pathways, higher frequency mutations and CNAs, relatively activated TME and increased ICMs expression, and better immunotherapeutic efficacy. While the high AIDPS group owned remarkably prolonged OS and RFS, a significant enrichment of metabolism-related pathways and mutational signature 3 (*BRCA1/2* mutations-related) was more sensitive to the ATPase inhibitor ouabain and HDAC inhibitor panobinostat. Therefore, in clinical settings, our AIDPS may become a reliable platform to further serve individualized decision-making in PACA.

In the era of precision medicine, anatomy-based TNM staging is far from meeting the needs of clinicians for ideal biomarkers that could accurately evaluate the prognosis and predict the efficacy of PACA patients (*Katz et al., 2008*; *Liu et al., 2022a*). Recently, numerous prognostic signatures of PACA have been constructed, but most of them are based on a specific biological pathway such as immunity, metabolic reprogramming, and m6A methylation (*Wang et al., 2022*; *Tan et al., 2020*; *Yuan et al., 2021*). This ignored information about other biological processes that played a crucial role in the oncogenesis and progression of PACA. In this study, based on 15,288 intersection genes in the training and nine testing cohorts, we further obtained 32 CPGs via univariate Cox regression analysis. In addition, in the existing studies, people mostly choose the modeling algorithms based on their knowledge limitations and preferences (*Liu et al., 2022a*; *Liu et al., 2022b*). To remedy this shortcoming, we collected 10 popular machine-learning algorithms that could be used to construct biomedical prognostic signatures. Among them, RSF, LASSO, CoxBoost, and Stepwise Cox have the function of variable screening and data dimensionality reduction, so we further combined them into 76 algorithm combinations. Ultimately, the combination of CoxBoost and Survival-SVM with the largest average C-index (0.675) in the remaining nine testing cohorts was identified as the final model.

Overfitting is one of the troublesome problems encountered by artificial intelligence and machine learning in biomedical model development, with several models fitting well in the training cohort but poorly in other external validation cohorts (*Deo, 2015*). After minimizing redundant information by CoxBoost, we finally obtained a nine-gene signature termed AIDPS through Survival-SVM. The results of Kaplan–Meier analysis, univariate Cox regression, ROC curve, and calibration curve all indicated that AIDPS had excellent predictive performance in the training cohort, nine testing cohorts, Meta-Cohort, and three external validation cohorts. Moreover, compared with 86 published PACA signatures, AIDPS exhibited distinctly superior accuracy than other models in almost all cohorts, revealing the robustness of AIDPS. It should be mentioned that, although the 20-gene signature of Demirkol CS, the 6-gene signature of Stratford JK, the 15-gene signature of Chen DT, and the 5-gene signature of Kim J, and so on were better than AIDPS in a certain cohort, but they were unsatisfactory across almost all validation cohorts, which might due to very poor generalization ability from overfitting. In contrast, it contains fewer elements and thus AIDPS has a superior extrapolation possibility, and the findings of 13 independent multicenter cohorts also fully confirm this notion.

In addition, compared with several common clinical and molecular characteristics such as TNM stage, grade, *KRAS*, *TP53*, or *CDKN2A* mutations, our AIDPS showed significantly improved accuracy. After stratifying PACA patients into the high and low AIDPS groups, we demonstrated that there was no outstanding difference in age, gender, and TNM stage, but the low AIDPS group had more advanced grades, which also contributed to its worse prognosis to some extent. Furthermore, given the excellent performance of AIDPS in PACA, we additionally measured its performance in four common digestive system tumors LIHC, STAD, COAD, and READ and found that AIDPS could accurately stratify patients. These findings indicated that AIDPS constructed in PACA, as a biomarker, has broad prospects for generalization to other tumors.

GSEA functional enrichment analysis was applied to elucidate the underlying biological mechanisms of AIDPS. The low AIDPS group was mostly enriched in immune and proliferation-related pathways, which partly explained its more advanced grades and worse prognosis. In addition, many recent studies have reported the emerging promise of epigenetic alterations especially DNA methylation in the early diagnosis and prognostic follow-up of PACA (*Yokoyama et al., 2020*; *Grady et al., 2021*). Therefore, we identified four MDGs in PACA patients. Further investigations found that higher methylation levels of *MAP3K8* and *PCDH7*, and lower methylation levels of *PCDHB1* and *SPAG6* in the high AIDPS, all corresponded to obviously prolonged OS, suggesting that methylation modification might play an indispensable role in its better prognosis.

Based on multi-omics data of TCGA-PAAD, we further investigated the genomic heterogeneity with regard to AIDPS. The results showed that the low AIDPS group owned higher TMB and superior mutation frequencies in the classical tumor suppressor genes *TP53*, *CDKN2A*, and oncogene *KRAS*. Numerous studies have revealed that *TP53*, *CDKN2A,* and *KRAS* mutations promoted invasion, metastasis, and immune escape in PACA patients, resulting in worse prognosis (*Hu et al., 2018*; *Hashimoto et al., 2019*). In addition, the results of CNA indicated that the low AIDPS group had evidently increased amplification of 8q24.21, 19q13.2, and 8p11.22 as well as deletion of 9p21.3, 18q21.2, 6p22.2, and 22q13.31 than those in the high AIDPS group. A recent study has shown that amplification of 19q13.2 and consequent overexpression of this loci was correlated with impaired survival in PACA (*Sandhu et al., 2016*). *Morikawa et al., 2018* also found that amplification of 8p11.22 was associated with chemotherapy resistance and shorter OS in ovarian clear cell carcinoma. *Baker et al., 2016* reviewed that people with 9p21.3 deletion were more susceptible to multiple types of tumors. Previous studies have demonstrated that deletion of 22q13.31 was an early genetic event in insulinoma independent of other genomic alteration and related to more advanced stage and dismal prognosis (*Jonkers et al., 2006*). Overall, all this evidence consistently supported our conclusion that genome-driven events might contribute to worse prognosis of the low AIDPS group. On the other hand, relatively high-frequency genomic alteration and higher TMB also provide more neoantigens, which might point the way for immunotherapy in the low AIDPS group (*Sha et al., 2020*).

Next, we investigated the immune landscape between the high and low AIDPS groups. The results of ssGSEA exhibited that the low AIDPS group possessed superior abundance of immune cell types, including activated CD4+ T cells, CD56dim natural killer cells, and central memory CD8+ T cells. As everyone knows, these increased effector cells will enhance antitumor immunity and bring better immunotherapeutic effects for the low AIDPS group (*Borst et al., 2018*; *Wu et al., 2013*). Over the past decade, ICIs targeting ICMs have shed light on the treatment of solid tumors (*Billan et al., 2020*). As expected, our findings showed that the expression of ICMs such as *CD274*, *CD276*, and *PDCD1LG2* was dramatically elevated in patients with low AIDPS, suggesting that they were more likely to benefit from ICIs treatment. TIDE and Submap are two widely recognized tools for predicting tumor patient sensitivity to ICIs based on expression profile (*Jiang et al., 2018*; *Hoshida et al., 2007*). Consistently, the results confirmed our previous conclusion that patients with low AIDPS possessed a greater response rate to immunotherapy. Overall, these findings indicated that our AIDPS might provide a reference for early identification of immunotherapy-sensitive PACA patients receiving first-line immunotherapy.

Precision medicine requires clinicians to identify patients who are sensitive to various treatments as early as possible for further individualized treatment. Therefore, considering the higher sensitivity of the low AIDPS group to immunotherapy, we integrated CTRP, PRISM, and CMap databases to develop specific drugs for patients in the high AIDPS group (*Yang et al., 2021*; *Subramanian et al., 2017*; *Malta et al., 2018*). Ultimately, an HDAC inhibitor, panobinostat, attracted our attention as a

potential drug for patients in the high AIDPS group. The latest studies report that panobinostat can synergistically enhance the antitumor effect of the selective Wee1 kinase inhibitor MK-1775 or CAR-T cell therapy in PACA (*Wang et al., 2015*; *Ali et al., 2021*). In the future, more clinical trials are required to confirm the broad prospects of panobinostat in PACA, especially in patients with high AIDPS.

After careful literature research, we found that *SELENBP1* that owned lower expression in the low AIDPS group with poor prognosis was associated with better tumor prognosis, and its decreased expression led to poor prognosis of many tumors such as lung adenocarcinoma (*Chen et al., 2004*). Meanwhile, *PRR11*, *EREG*, *ADM,* and *TGM2*, which were overexpressed in the low AIDPS group, played a vital role in the proliferation, invasion, and metastasis of many tumors such as breast cancer, colorectal cancer, and PACA, and lead to their chemotherapy resistance and were potential targets for enhanced efficacy (*Lee et al., 2020*; *Neufert et al., 2013*; *Aggarwal et al., 2012*; *Malkomes et al., 2021*). In addition, our study also found strong biological correlations among nine AIDPS genes, which are also closely related to mutations, CNAs, and TME in PACA. This also essentially supports the important impact of AIDPS on the prognosis and immune microenvironment, as well as its role in the clinical management and precision treatment of PACA.

This study differed from previous studies in the following aspects. (1) We systematically collected 13 large multicenter cohorts and selected the algorithm with the largest average C-index in the 9 testing cohorts to construct our AIDPS. (2) Unlike current prognostic models for a certain pathway, our AIDPS was based on 15,288 intersection genes from training and nine testing cohorts, which avoided the omission of other indispensable biological process in the initiation and progression of PACA. (3) In order to prevent the inappropriate modeling methods due to personal preference, we combined 10 recognized machine-learning algorithms into 76 combinations and selected the best model based on their accuracy. While we have tried to be as rigorous and comprehensive as possible in our research, some limitations should be noted. First, although we collected 13 independent multicenter cohorts, further validation in prospective study was warranted. Second, in spite of the nine genes included in AIDPS having appeared in numerous prognostic signatures of PACA, which indicated their consistent prognostic value. Their roles in PACA remain to be elucidated, and more functional experimental validation is required in the future. Finally, further clinical trials are necessary to affirm the therapeutic effect of panobinostat in PACA patients with high AIDPS.

In conclusion, based on 32 CPGs from training cohort, nine testing cohorts, and three external validation cohorts, we constructed and validated a consensus prognostic signature (termed AIDPS) via 76 machine-learning algorithm combinations. After incorporating several vital clinicopathological features and 86 published signatures, AIDPS also exhibited robust and dramatically superior predictive capability. Of note, our AIDPS has important clinical implications for the clinical management and individualized treatment of PACA, and patients with low AIDPS are more sensitive to immunotherapy, while panobinostat may be a potential agent for patients with high AIDPS. In addition, in other prevalent digestive system tumors, the nine-gene AIDPS could still accurately stratify the prognosis, suggesting a strong possibility of extrapolation. Overall, our study provides an attractive tool for prognostic evaluation, risk stratification, and individualized treatment of PACA patients in clinical practice.

## Materials and methods
### Data acquisition and preprocessing
We collected datasets from The Cancer Genome Atlas (TCGA, http://portal.gdc.cancer.gov/), International Cancer Genome Consortium (ICGC, http://dcc.icgc.org/), ArrayExpress (https://www.ebi.ac.uk/arrayexpress/), and Gene Expression Omnibus (GEO, https://www.ncbi.nlm.nih.gov/geo/) public databases according to the following procedure: (1) more than 40 samples with survival information; (2) at least 15,000 clearly annotated genes; and (3) patients with primary tumors and no other treatments were given before resection. Finally, we enrolled 1570 samples from 13 cohorts, TCGA-PAAD (n = 176), ICGC-PACA-AU-Seq (PACA-AU-Seq, n = 81), ICGC-PACA-AU-Array (PACA-AU-Array, n = 267), ICGC-PACA-CA (PACA-CA-Seq, n = 182), E-MTAB-6134 (n = 288), GSE62452 (n = 65), GSE28735 (n = 42), GSE78229 (n = 49), GSE79668 (n = 51), GSE85916 (n = 79), GSE21501 (n = 102), GSE57495 (n = 63), and GSE71729 (n = 125). The FPKM data in the TCGA-PAAD was downloaded from UCSC Xena database (https://xenabrowser.net/datapages/) and further converted into log2 (TPM + 1) format. The

RNA-Seq data of ICGC were downloaded from its portal and converted into log2 (TPM + 1) format. The normalized exp-Array data from ICGC, ArrayExpress, and GEO were generated directly from their portal. Detailed clinical and pathological information of these 13 cohorts is presented in *Figure 1—source data 1*.

### Univariate Cox regression analysis

Based on intersection genes, we performed univariate Cox regression analysis in the training cohort, nine testing cohorts. We selected consensus prognosis genes (CPGs) for the next study according to the following criteria: p<0.05 and all hazard ratios (HRs) consistently >1 or <1 in more than 8/10 cohorts.

### Artificial intelligence-derived prognostic signature

To construct a consensus prognosis model for PACA, we performed our previous workflow (*Liu et al., 2022a*; *Liu et al., 2022b*). (1) First, we integrated 10 classical algorithms: random forest (RSF), least absolute shrinkage and selection operator (LASSO), gradient boosting machine (GBM), survival support vector machine (Survival-SVM), supervised principal components (SuperPC), ridge regression, partial least squares regression for Cox (plsRcox), CoxBoost, Stepwise Cox, and elastic network (Enet). Among them, RSF, LASSO, CoxBoost, and Stepwise Cox have the function of dimensionality reduction and variable screening, and we combined them with other algorithms into 76 machine-learning algorithm combinations. (2) Next, we utilized the PACA-AU-Array with larger sample size in ICGC as the training cohort and used these 76 combinations to construct signatures separately in the expression files with 32 CPGs. (3) Finally, in the nine testing cohorts (TCGA-PAAD, PACA-AU-Seq, PACA-CA-Seq, E-MTAB-6134, GSE62452, GSE28735, GSE78229, GSE79668, GSE85916), we calculated the AIDPS score for each cohort using the signature obtained in the training cohort. Based on the average C-index of the nine testing cohorts, we finally picked the best consensus prognosis model for PACA.

### Validating the prognostic value of AIDPS in 14 datasets

Patients in the training cohort, nine testing cohorts, Meta-Cohort (obtained by removing batch effects and repetitions from training cohort and nine testing cohorts), and three external validation cohorts (GSE21501, GSE57495, and GSE71729) were categorized into high and low AIDPS groups according to the median value. The prognostic value of AIDPS was evaluated by Kaplan–Meier curve and multivariate Cox regression analysis. The calibration curve and receiver-operator characteristic (ROC) curve were plotted to assess the predictive accuracy of AIDPS.

### Collection and calculation of PACA published signatures

With the attention paid to the stratified management and precise treatment of PACA, many prognostic signatures have been constructed, including m6A-related lncRNA signature, metabolic reprogramming-related signature, and SMAD4-driven immune signature, etc. (*Wang et al., 2022*; *Tan et al., 2020*; *Yuan et al., 2021*). To compare the predictive performance of AIDPS and these published signatures, we systematically searched PubMed for published prognostic model articles up to January 1, 2022. Afterward, we calculated their risk scores in the 14 cohorts based on the genes and coefficients provided by the article, and comprehensively evaluated their prognostic performance in PACA by univariate Cox analysis and C-index.

### Evaluating clinical significance of AIDPS

We compared the differences in several pivotal clinical traits such as age, gender, TNM stage, and grade between the high and low AIDPS groups. In addition, to explore the application value of AIDPS in other prevalent gastrointestinal tumors, we acquired the mRNA expression and survival data of LIHC, STAD, COAD, and READ in the same way as TCGA-PAAD, and further performed Kaplan–Meier analysis.

### Gene set enrichment analysis

GSEA was applied to identify specific functional pathways in the high and low AIDPS groups. After differential analysis using the *DESeq2* package, all genes were ranked in descending order according to log2FoldChange (log2FC). Next, we identified GO and KEGG enriched pathways by the

*clusterProfiler* package and further selected the top five pathways in normalized enrichment score (NES) for visualization.

## Genomic alteration landscape

To investigate the genomic alteration landscape in the high and low AIDPS groups, we performed a comprehensive analysis of mutation and CNA data in the TCGA-PAAD. (1) After obtaining the raw mutation file, we calculated the TMB of each sample and visualized the top 15 genes through the *maftools* package. (2) As described in *Lu et al., 2021*, we applied the *deconstructSigs* package to extract the mutational signatures for each PACA patients, and selected mutational signature 1 (age-related), mutational signature 2 (*APOBEC* activity-related), mutational signature 3 (*BRCA1/2* mutations-related), and mutational signature 6 (dMMR-related) with higher frequency of occurrence in PACA for visualization (*Alexandrov et al., 2013*). (3) Recurrent amplified or deleted genome regions were decoded and localized through GISTIC 2.0 module in Firebrowse tool (http://firebrowse.org/). We finally selected regions with broad-level CNA frequency >20% and several genes located within chromosomes 8q24.21, 9p21.3, and 18q21.2 for display.

## Estimation of methylation-driven events

Following the pipeline developed in previous studies (*Liu et al., 2021b*; *Liu et al., 2021a*), we identified MDGs for TCGA-PAAD. Furthermore, we compared the differences in the methylation levels and mRNA expression levels of MDGs in the two groups, and further evaluated the effect of MDGs methylation levels on the prognosis by Kaplan–Meier survival curve.

## Immune molecule expression and tumor microenvironment evaluation

The ssGSEA was utilized to comprehensively infer the infiltration abundance of immune and stromal component in the high and low AIDPS groups (*Wang et al., 2022*; *Liu et al., 2021c*). In addition, we recruited 27 ICMs from our previous study, including *B7-CD28* family, *TNF* superfamily, etc., and measured their expression levels between the two groups (*Wang et al., 2022*).

## Response to immunotherapy

TIDE web tool was used to predict responsiveness to ICIs between the high and low AIDPS groups, and lower TIDE scores suggested better immunotherapeutic efficacy (*Jiang et al., 2018*). Additionally, we applied the Submap to calculate the expression similarity between patients in the high and low AIDPS groups and patients who responded/non-responded to ICIs, and then speculated immunotherapy efficacy (*Hoshida et al., 2007*).

## Development and validation of potential therapeutic agents

As shown in *Figure 8D*, we developed potential therapeutic agents for the high AIDPS group according to the protocol of *Yang et al., 2021*. (1) First, we acquired drug sensitivity data for CCLs from the CTRP as well as PRISM repurposing datasets, and expression data of CCLs from the Cancer Cell Line Encyclopedia (CCLE) database. (2) The CTRP and PRISM datasets own AUC values, and lower AUC values suggest increased sensitivity to this compound. Moreover, as a first-line chemotherapeutic drug for PACA, we further selected gemcitabine to verify the scientific and rigor of this approach. (3) Based on Wilcoxon rank-sum test, we performed a differential analysis of drug response between the high AIDPS (first 10%) and low AIDPS (last 10%) groups, and the threshold log2FC > 0.2 was set to identify compounds with lower AUC values in the high AIDPS group. (4) Next, we applied Spearman correlation to further screen compounds with AUC values that had negative correlation coefficients with AIDPS (setting threshold $R<-0.4$). (5) Finally, we identified potential drugs for patients in the high AIDPS group by the intersection of the compounds obtained from (3) and (4).

The CMap (https://clue.io/) is a publicly available web tool for discovering candidate compounds that may target AIDPS-related pathways based on gene expression signature (*Subramanian et al., 2017*; *Malta et al., 2018*). Based on differential expression analysis, we identified potential compounds in PACA using CMap to further validate the results obtained from the CTRP and PRISM databases.

## Statistical analysis

All data cleaning, analysis, and result visualization for this research were performed in R 4.1.2. Continuous variables were analyzed by Wilcoxon rank-sum test or Student's *t*-test. Categorical variables

were statistically compared using Chi-square test or Fisher's exact test. The *survival* package was used for univariate, multivariate Cox as well as Kaplan–Meier survival analysis. The *timeROC* package and *rms* package were utilized to plot ROC curve and calibration curve, respectively. The *MethylMix* package was applied to identify MDGs. p-Value (two-sided) < 0.05 was considered statistically significant.

## Data availability

All data generated during this study are included in the manuscript and supporting files. The basic script for our AIDPS model is available on the GitHub website (https://github.com/Zaoqu-Liu/AIDPS, copy archived at swh:1:rev:f9d929456fa9f0cd7721cea6a5e2a1412ffae4a7; *Liu, 2022*); the researchers entered the expression matrix of these nine AIDPS genes and could obtain patient-specific risk scores. Source data files have been provided for *Figures 1, 2 and 4*.

## Acknowledgements

We sincerely thank the research group that contributed pancreatic cancer sequencing data as well as the staff who developed the R package. This study was supported by the National Natural Science Foundation of China (nos. 81870457, 82172944).

---

## Additional information

### Funding

| Funder | Grant reference number | Author |
| --- | --- | --- |
| National Natural Science Foundation of China | 81870457 | Yu-ling Sun |
| National Natural Science Foundation of China | 82172944 | Yu-ling Sun |

The funders had no role in study design, data collection and interpretation, or the decision to submit the work for publication.

### Author contributions

Libo Wang, Data curation, Formal analysis, Validation, Investigation, Visualization, Writing – original draft; Zaoqu Liu, Conceptualization, Resources, Software, Methodology, Project administration, Writing – review and editing; Ruopeng Liang, Validation, Writing – review and editing; Weijie Wang, Rongtao Zhu, Jian Li, Writing – review and editing; Zhe Xing, Siyuan Weng, Resources, Data curation, Writing – review and editing; Xinwei Han, Supervision, Writing – review and editing; Yu-ling Sun, Supervision, Funding acquisition, Writing – review and editing

### Author ORCIDs

Libo Wang http://orcid.org/0000-0003-3745-9459
Zaoqu Liu http://orcid.org/0000-0002-0452-742X
Xinwei Han http://orcid.org/0000-0003-4407-4864
Yu-ling Sun http://orcid.org/0000-0001-5289-4673

### Decision letter and Author response

Decision letter https://doi.org/10.7554/eLife.80150.sa1
Author response https://doi.org/10.7554/eLife.80150.sa2

---

## Additional files

### Supplementary files
• MDAR checklist

## Data availability

Figure 1-source data 1, Figure 2-source data 1 and Figure 4-source data 1 contain the numerical data used to generate the figures.

The following previously published datasets were used:

| Author(s) | Year | Dataset title | Dataset URL | Database and Identifier |
|---|---|---|---|---|
| Goldman MJ, Craft B, Hastie M | 2020 | GDC TCGA Pancreatic Cancer (PAAD) | https://xenabrowser.net/datapages/?cohort=GDC%20TCGA%20Pancrea | The Cancer Genome Atlas, TCGA-PAAD |
| Zhang J, Bajari R, Andric D | 2019 | exp_seq. PACA-AU | https://dcc.icgc.org/api/v1/download?fn=/current/Projects/PACA-AU/exp_seq.PACA-AU.tsv.gz | International Cancer Genome Consortium, PACA-AU-Seq |
| Zhang J, Bajari R, Andric D | 2019 | exp_array. PACA-AU | https://dcc.icgc.org/api/v1/download?fn=/current/Projects/PACA-AU/exp_array.PACA-AU.tsv.gz | International Cancer Genome Consortium, PACA-AU-Array |
| Zhang J, Bajari R, Andric D | 2019 | exp_seq. PACA-CA | https://dcc.icgc.org/releases/current/Projects/PACA-CA | International Cancer Genome Consortium, PACA-CA-Seq |
| Puleo F, Nicolle R, Blum Y | 2018 | mRNA profiling by array for pancreatic ductal adenocarcinoma for clinical application | https://www.ebi.ac.uk/arrayexpress/experiments/E-MTAB-6134/ | ArrayExpress, E-MTAB-6134 |
| Yang S, He P, Wang J | 2016 | Microarray gene-expression profiles of 69 pancreatic tumors and 61 adjacent non-tumor tissue from patients with pancreatic ductal adenocarcinoma | https://www.ncbi.nlm.nih.gov/geo/query/acc.cgi?acc=GSE62452 | NCBI Gene Expression Omnibus, GSE62452 |
| Zhang G, He P, Tan H | 2012 | Microarray gene-expression profiles of 45 matching pairs of pancreatic tumor and adjacent non-tumor tissues from 45 patients with pancreatic ductal adenocarcinoma | https://www.ncbi.nlm.nih.gov/geo/query/acc.cgi?acc=GSE28735 | NCBI Gene Expression Omnibus, GSE28735 |
| Wang J, Yang S, He P | 2017 | Microarray gene-expression profiles of 50 pancreatic tumors tissue from patients with pancreatic ductal adenocarcinoma | https://www.ncbi.nlm.nih.gov/geo/query/acc.cgi?acc=GSE78229 | NCBI Gene Expression Omnibus, GSE78229 |
| Kirby MK, Ramaker RC | 2016 | RNA-sequencing of human pancreatic adenocarcinoma cancer tissues | https://www.ncbi.nlm.nih.gov/geo/query/acc.cgi?acc=GSE79668 | NCBI Gene Expression Omnibus, GSE79668 |
| Puleo F, Maréchal R, Demetter P | 2018 | Patients with human resected pancreatic cancer | https://www.ncbi.nlm.nih.gov/geo/query/acc.cgi?acc=GSE85916 | NCBI Gene Expression Omnibus, GSE85916 |
| Stratford JK, Yeh JJ | 2010 | A six-gene signature predicts survival of patients with localized pancreatic ductal adenocarcinoma | https://www.ncbi.nlm.nih.gov/geo/query/acc.cgi?acc=GSE21501 | NCBI Gene Expression Omnibus, GSE21501 |
| Chen D, Malafa MP | 2015 | Development of a prognostic gene signature in pancreatic cancer | https://www.ncbi.nlm.nih.gov/geo/query/acc.cgi?acc=GSE57495 | NCBI Gene Expression Omnibus, GSE57495 |

*Continued on next page*

*Continued*

| Author(s) | Year | Dataset title | Dataset URL | Database and Identifier |
|-----------|------|---------------|-------------|-------------------------|
| Jen Jen Y | 2015 | Virtual Microdissection of Pancreatic Ductal Adenocarcinoma Reveals Tumor and Stroma Subtypes | https://www.ncbi.nlm.nih.gov/geo/query/acc.cgi?acc=GSE71729 | NCBI Gene Expression Omnibus, GSE71729 |

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

## Appendix 1

### Current research progress on the nine AIDPS genes

The nine AIDPS genes were rarely studied in tumors, most of which focused on their roles in biological phenotypes such as tumorigenesis and progression, and a few studies investigated their roles and guiding significance in PACA immune microenvironment and treatment. Specifically, we found that SELENBP1-owned lower expression in the low AIDPS group with poor prognosis was associated with better tumor prognosis, and its decreased expression resulted in poor prognosis of many tumors such as lung adenocarcinoma. Meanwhile, DCBLD2, PRR11, EREG, ADM, and TGM2, which were overexpressed in the low AIDPS group, played an essential role in the proliferation, invasion, and metastasis of many tumors such as PACA, hepatocellular carcinoma, and colorectal cancer, and lead to their chemotherapy resistance and were potential targets for enhanced efficacy. However, the functional roles and mechanisms of UNC13D and CDCA4 in tumors are not well understood, and more high-quality studies are needed in the future. These findings are consistent with our study, and the specific functional characteristics are summarized as follows:

1. SELENBP1, encoding a novel human methanethiol oxidase, whose expression is significantly decreased in the progression of Barrett's esophagus to adenocarcinoma and can affect chemotherapy sensitivity (*Silvers et al., 2010*). This is consistent with our results that SELENBP1 expression was decreased in the low AIDPS group with worse prognosis (*Figure 7—figure supplement 3*).

2. PLCB4 encodes a phospholipase, and activating PLCB4 mutation was recently identified in primary leptomeningeal melanocytic tumors (*Van de nes et al., 2017*).

3. DCBLD2 is upregulated in glioblastoma and head and neck cancer, and phosphorylation of DCBLD2 Y750 activates AKT pathway, which in turn enhances EGFR-driven tumorigenesis (*Feng et al., 2014*). DCBLD2 has also been found to play a crucial role in the invasion, progression, and metastasis of neuroendocrine tumors (*Hofsli et al., 2008*). These results are consistent with an increased expression of DCBLD2 in the low AIDPS group with a dismal prognosis.

4. Proline-rich protein 11 (PRR11) is a novel tumor-related gene that has been found to promote breast cancer growth and anti-estrogen resistance through the PI3K pathway (*Lee et al., 2020*). In addition, it has also been demonstrated that PRR11 affects autophagy and promotes proliferation in non-small cell lung cancer (NSCLC) through the Akt/mTOR pathway (*Zhang et al., 2018*; *Wei et al., 2022*), and recent studies have also found that PRR11 can promote the tumorigenesis and progression of renal clear cell carcinoma (ccRCC) by regulating E2F1 stability (*Chen et al., 2021*). Overall, these results are consistent with higher expression of PRR11 in the low AIDPS group with a worse prognosis.

5. UNC13D mutation plays an essential role in familial hemophagocytic lymphohistiocytosis (FHL), but its role in tumors remains unknown and needs to be further explored (*Meeths et al., 2011*; *Cichocki et al., 2014*; *Spessott et al., 2015*; *Borte et al., 2014*).

6. EREG, as an EGFR ligand epithelial regulatory protein, plays a fundamental role in the growth and proliferation of breast and colorectal cancers (*Farooqui et al., 2015*; *Neufert et al., 2013*) and is associated with tamoxifen resistance in breast cancer (*He et al., 2019*) and efficacy of panitumumab and adjuvant chemotherapy in colorectal cancer (*Smyth et al., 2016*; *Thomaidis et al., 2014*).

7. Adrenomedullin (ADM) encodes a pre-prohormone that is cleaved into two bioactive peptides, which can dilate blood vessels, regulate hormone secretion, and promote angiogenesis. Many studies have confirmed that ADM plays a major role in the proliferation, invasion, angiogenesis, and metastasis of PACA, hepatocellular carcinoma, prostate cancer, melanoma, ovarian cancer, and endometrial cancer and may become a novel therapeutic target (*Aggarwal et al., 2012*; *Ramachandran et al., 2007*; *Park et al., 2008*; *Berenguer-Daizé et al., 2013*; *Chen et al., 2011*; *Deng et al., 2012*; *Oehler et al., 2001*).

8. In addition, currently one of the few studies supports the oncogenic role of cell division cycle-associated protein 4 (CDCA4) in tumors and nuclear factors induced as E2F transcription factor families that regulate E2F-dependent transcriptional activation and cell proliferation, but their potential mechanism, especially effects on PACA tumorigenesis and progression, requires further investigation (*Fang et al., 2022*; *Hayashi et al., 2006*).

9. Transglutaminase 2 (TGM2), a promoter of stemness and radiotherapy resistance in glioma (*Berg et al., 2021*), has been confirmed in many studies to promote colorectal cancer progression via P53 and Wnt/β-catenin signaling pathways (*Malkomes et al., 2021*; *Gu et al., 2019*), as well as is associated with tumor-promoting inflammation in gastric cancer (*Cho et al., 2020*). These results are consistent with the TGM2 expression dramatically increased in the low AIDPS group with worse prognosis, suggesting that TGM2 may play an important role in the development and progression of PACA.

