## [Editor Report]

This work sets out to develop a better machine learning-based predictor of survival/prognosis for patients diagnosed with pancreatic cancer. To achieve this, the authors developed a large combinatorial family of machine learning methods based on a high-dimensional set of -omics and other patient data features; using ten publicly available data sets. By finding the combined ML method(s) that performed best on the task, the authors were able to identify a reduced set of features (giving rise to a signature called AIDPS that involves 9 genes) which, when measured in the patient, allow for fairly accurate prediction of patient survival and prognosis. Importantly, three new external data sets GSE21501, GSE57495, GSE71729 were used in the validation.

---

## [Decision Letter]

**Decision letter after peer review:**

Thank you for submitting your article "Comprehensive machine-learning survival framework develop a consensus model in large scale multi-center cohorts for pancreatic cancer" for consideration by *eLife*. Your article has been reviewed by 3 peer reviewers, and the evaluation has been overseen by Caigang Liu as the Senior Editor. The following individual involved in the review of your submission has agreed to reveal their identity: Anthony C C Coolen (Reviewer #2).

Essential revisions:

1. The authors should repeat their complete pipeline after randomizing within each of their ten data sets the clinical outcomes (i.e. shuffle the time-to-event outcomes). If one would obtain again a signature with comparable prognostic features, then one can be sure that what is reported here is largely a manifestation of overfitting due to dimension mismatch, and hence of no interest.

2. The authors should provide a better understanding of why agreement/optimization amongst a large number of combined ML methods was chosen as a way to drive feature selection. Comparing to existing methods is one thing, currently, there isn't really a clear comparison here to other feature selection approaches.

3. The authors should provide a thorough explanation of the biological relevance of the 9 genes in relation to the mutational load/copy number changes/ immunological infiltration found in low AIDPS patients.

*Reviewer #1 (Recommendations for the authors):*

Things I think would strengthen the paper:

1. A better understanding of why agreement/optimization amongst a large number of combined ML methods was chosen as a way to drive feature selection. Comparing to existing methods is one thing, but there isn't really a clear comparison here to other feature selection approaches, nor much discussion of why this one was chosen. Is this meant to be a reproducible approach, for other problems, or is the point just to get the AIDPS features? Why is this approach to feature selection preferred/superior.

2. Right now the current pipeline seems highly unwieldy because of all the models considered. Is this meant to be a general and reproducible approach? if so, the authors need to demonstrate more clearly that this same approach can be done by others with ease.

3. Everything in the subsequent analyses comparing high and low AIDPS groups hinges on the assumed accuracy and generality of the model. Once you have 9 genes, though, you should be able to study those features using a range of methods, some of which are picked to have high interpretability. I am sorry if I missed this, but at the moment it is not clear to me what best an ML method can do is using the 9 genes, and whether there is an interpretable method that works well but might imply a non-trivial relationship amongst the genes that matters to the outcome. This seems more informative than studying the statistics of high and low AIDPS groups

*Reviewer #2 (Recommendations for the authors):*

To rescue their claims one could suggest to the authors the following simple but clean and fair test. The authors could repeat their complete pipeline after randomizing within each of their ten data sets the clinical outcomes (i.e. shuffle the time-to-event outcomes). If that exercise is found to make the signal disappear (i.e. the separation of KM curves, and the nontrivial AUROC values), then their present results should be studied further as possibly relevant. If, on the contrary, one would obtain again a signature with comparable prognostic features, then one can be sure that what is reported here is largely a manifestation of overfitting due to dimension mismatch, and hence of no interest.

*Reviewer #3 (Recommendation for the authors):*

From the biological point of view, currently, there was little explanation of the biological relevance of the 9 genes in relation to the mutational load/copy number changes/ immunological infiltration found in low AIDPS patients. Please discuss these in sufficient detail.

[Editors' note: further revisions were suggested prior to acceptance, as described below.]

Thank you for resubmitting your work entitled "Comprehensive machine-learning survival framework develop a consensus model in large scale multi-center cohorts for pancreatic cancer" for further consideration by *eLife*. Your revised article has been evaluated by Caigang Liu (Senior Editor) and a Reviewing Editor.

The manuscript has been improved but there are some remaining issues that need to be addressed, as outlined below:

1. The improvements to figure 7 (in the supplementary figures linked to Figure 7) meant to address Reviewer #1's request seem to all focus on treating the aidps features separately. Please show the accuracy when these aidps features are considered together as a nine dimensional feature vector in one model.

2. For reviewer #2, please remove the newly added material called `performance of AIDPS in randomly split cohorts', as this adds nothing relevant (as it s again testing on discovery data). Please take out all mentioning of `validation' or `validated' from the manuscript, other than when discussing the tests on the three new data sets GSE21501, GSE57495, GSE71729; only the latter can be called validation sets. Please compare the performance of existing signatures to that of the authors' own signature on the three new data sets GSE21501, GSE57495, GSE71729.

The main reason for this suggestion is that the revised manuscript still tests their own signature on discovery data but that of the competitors on unseen data, which is not a fair comparison. Unless the authors remedy this they should (i) remove all mentioning of superior performance of their signature (since the comparison did not involve a level paying field) and (ii) remove all mentioning of `validation' other than in relation to the three new data sets.

3. Please include (4. Current research progress on these nine AIDPS genes) in the rebuttal, which is a summary of what is known about the biological function of the nine genes, as an Appendix or Supplementary information in the manuscript.

*Reviewer #1 (Recommendations for the authors):*

The authors have largely addressed the concerns raised in my previous review. As best I can tell, they have not yet shown how a simple ML model (such as linear regression or xgboost) performs on the same task when only fed with the 9 aidps features simultaneously. The improvements to figure 7 meant to address my request seem to all focus on treating the aidps features separately, so it useful to know they perform poorly when separate… what about when they are considered together as a nine dimensional feature vector in one model?

*Reviewer #2 (Recommendations for the authors):*

In my first report I put forward two points. First, that the authors used the same data sets both in the discovery stage and the validation stage. Second, that they compared the performance of competitor signatures to theirs by using these same data. This has the following consequences: 1. They could not claim that their signature had been validated, because what they called validation data were in fact the discovery data (the potential impact of overfitting is hence unclear). 2. The comparison between their signature and the existing ones was not fair (their own signature was tested on its corresponding discovery data, but the competitors' signatures on what for those signatures would be independent validation data). Therefore, all claims on their signature having been validated and on it being superior to existing signatures were not supported by the available evidence. I also suggested that they could test for overfitting impact by repeating their previous analyses step by step with outcome-randomized versions of the original data. This should make all true signals disappear and is hence a clean and fair test.

In the revised version of their manuscript the authors did not carry out the requested re-analysis with outcome-randomized data. Instead they (a) sampled several new data sets from their original data (without outcome randomization), and (b) tested their signature on three independent new data sets. Exercise (a) seems pointless; of course the signature would again be able to predict, since one is testing it once more on samples from the discovery set. Exercise (b), although not what was requested by this referee, is more useful. Here the authors report again significant performance of their signature, with AUC values in the range 0.68-0.70, lending weight to the suggestion that their signature makes sense.

I conclude from the authors' reply and from the new version of the paper that point 1. above has been partly addressed, but not point 2. If the authors, as it seems, do not wish to carry out the requested analysis with outcome randomization, then they have the following options:

Option A;

– Remove the newly added material called `performance of AIDPS in randomly split cohorts', this adds nothing relevant (see my notes above, it is again testing performance on discovery data).

– Take out all mentioning of `validation' or `validated' from the manuscript, other than when discussing the tests on the three new data sets GSE21501, GSE57495, GSE71729; only the latter can be called validation sets.

– Remove all performance comparisons between the new signature and the existing ones (since this comparison is presently not fair) and all conclusions based on these comparisons.

Option B:

– Remove the newly added material called `performance of AIDPS in randomly split cohorts', this adds nothing relevant (see my notes above, it is again testing on discovery data).

– Take out all mentioning of `validation' or `validated' from the manuscript, other than when discussing the tests on the three new data sets GSE21501, GSE57495, GSE71729; only the latter can be called validation sets.

– Compare the performance of existing signatures to that of their own signature on the three new data sets GSE21501, GSE57495, GSE71729; this comparison would be fair.

I would personally strongly suggest option B, since otherwise the paper would in my view be too weak in terms of the amount of justified conclusions to warrant publication (but that is a decision for the editor to make). In addition, I would still hope that the authors choose to comply with my original request for application of the full pipeline to outcome-randomized discovery data. This would clarify objectively, for the authors' own benefit, the extent to which their original tests and conclusions may have been affected by overfitting.

*Reviewer #3 (Recommendations for the authors):*

The authors have addressed my comments. I find the following v useful as a summary of what is known about the biological function of the nine genes and wonder if it can be incorporated as an Appendix or Supplementary information:

"The nine AIDPS genes were rarely studied in tumors, most of which focused on their roles in biological phenotypes such as tumorigenesis and progression, and few studies investigated their roles and guiding significance in PACA immune microenvironment and treatment. Specifically, we found that SELENBP1 owned lower expression in the low AIDPS group with poor prognosis was associated with better tumor prognosis, and its decreased expression resulted in poor prognosis of many tumors such as lung adenocarcinoma. Meanwhile, DCBLD2, PRR11, EREG, ADM and TGM2, which were overexpressed in the low AIDPS group, played an essential role in the proliferation, invasion and metastasis of many tumors such as PACA, hepatocellular carcinoma and colorectal cancer, and lead to their chemotherapy resistance and were potential targets for enhanced efficacy. However, the functional roles and mechanisms of UNC13D and CDCA4 in tumors are not well understood, and more high-quality studies are needed in the future. These findings are consistent with our study, and the specific functional characteristics are summarized as follows:

1. SELENBP1, encoding a novel human methanethiol oxidase, whose expression is significantly decreased in the progression of Barrett's esophagus to adenocarcinoma and can affect chemotherapy sensitivity (PMID: 20332323). This is consistent with our results that SELENBP1 expression was decreased in the low AIDPS group with worse prognosis (Figure 7—figure supplement 3).

2. PLCB4 encodes a phospholipase, and activating PLCB4 mutation was recently identified in primary leptomeningeal melanocytic tumors (PMID: 28499758).

3. DCBLD2 is upregulated in glioblastoma and head and neck cancer, and phosphorylation of DCBLD2 Y750 activates AKT pathway which in turn enhances EGFR-driven tumorigenesis (PMID: 25061874). DCBLD2 has also been found to play a crucial role in the invasion, progression, and metastasis of neuroendocrine tumors (PMID: 18827820). These results are consistent with an increased expression of DCBLD2 in the low AIDPS group with a dismal prognosis.

4. Proline-rich protein 11 (PRR11) is a novel tumor-related gene, which has been found can promote breast cancer growth and anti-estrogen resistance through the PI3K pathway (PMID: 33127913). In addition, it has also been demonstrated that PRR11 affects autophagy and promotes proliferation in non-small cell lung cancer (NSCLC) through the Akt/mTOR pathway (PMID: 30258945; 35005120), and recent studies have also found that PRR11 can promote the tumorigenesis and progression of renal clear cell carcinoma (ccRCC) by regulating E2F1 stability (PMID: 34499617). Overall, these results are consistent with higher expression of PRR11 in the low AIDPS group with a worse prognosis.

5. UNC13D mutation play an essential role in familial hemophagocytic lymphohistiocytosis (FHL), but its role in tumors remains unknown and needs to be further explored (PMID: 21931115; 24842371; 25564401; 24882743).

6. EREG, as an EGFR ligand epithelial regulatory protein, plays a fundamental role in the growth and proliferation of breast and colorectal cancers (PMID: 26215578; 23549083) and is associated with tamoxifen resistance in breast cancer (PMID: 30967627) and efficacy of panitumumab and adjuvant chemotherapy in colorectal cancer (PMID: 26869404; 25272957).

7. Adrenomedullin (ADM) encodes a pre-prohormone that is cleaved into two bioactive peptides, which can dilate blood vessels, regulate hormone secretion, and promoting angiogenesis. Many studies have confirmed that ADM plays a major role in the proliferation, invasion, angiogenesis, and metastasis of PACA, hepatocellular carcinoma, prostate cancer, melanoma, ovarian cancer, and endometrial cancer and may become a novel therapeutic target (PMID: 22960655; 17363587; 18657357; 24100627; 21994414; 22400488; 11420706).

8. In addition, currently one of the few studies support the oncogenic role of cell division cycle associated protein 4 (CDCA4) in tumors and nuclear factors induced as E2F transcription factor families that regulate E2F-dependent transcriptional activation and cell proliferation, but their potential mechanism, especially effect on PACA tumorigenesis and progression require further investigation (PMID: 35251007; 16984923).

9. Transglutaminase 2 (TGM2), a promoter of stemness and radiotherapy resistance in glioma (PMID: 33483373), has been confirmed in many studies to promote colorectal cancer progression via P53 and Wnt/β-catenin signaling pathways (PMID: 34103685; 31570702), as well as is associated with tumor-promoting inflammation in gastric cancer (PMID: 32467608). These results are consistent with the TGM2 expression dramatically increased in the low AIDPS group with worse prognosis, suggesting that TGM2 may play an important role in the development and progression of PACA."

---

## [Author Response]

Essential revisions:1. The authors should repeat their complete pipeline after randomizing within each of their ten data sets the clinical outcomes (i.e. shuffle the time-to-event outcomes). If one would obtain again a signature with comparable prognostic features, then one can be sure that what is reported here is largely a manifestation of overfitting due to dimension mismatch, and hence of no interest.

Thank you and reviewer 2 for your insightful recommendations. In response to this problem, we supplement the contents of Figure 3—figure supplement 4-5, and detailed explanations are given in lines 584-637 of this author’s response. The results from 22 randomized split validation cohorts and three external test cohorts consistently confirmed that AIDPS can robustly predict the prognosis of pancreatic cancer (PACA) patients. Thanks again for your advice, which we are sure makes our validation of AIDPS more rigorous, reliable and meaningful.

2. The authors should provide a better understanding of why agreement/optimization amongst a large number of combined ML methods was chosen as a way to drive feature selection. Comparing to existing methods is one thing, currently, there isn't really a clear comparison here to other feature selection approaches.

Thank you and reviewer 1 for your questions. As to why we choose agreement/optimization as our feature selection method, we conducted in-depth thinking and careful investigation again, and made a detailed and rigorous discussion. Specific responses are presented in the recommendation section of reviewer 1 in lines 120-213.

3. The authors should provide a thorough explanation of the biological relevance of the 9 genes in relation to the mutational load/copy number changes/ immunological infiltration found in low AIDPS patients.

Thank you and reviewer 3 for your recommendations. As you mentioned, our study explored the genomic landscape such as mutation and copy number alteration (CNA) of the 9-gene AIDPS and their impact on the tumor microenvironment (TME), focusing on its guiding value for decision-making and clinical management in PACA patients, while ignoring the biological functions of these nine genes as independent individuals. To compensate for this defect, we supplemented two parts, Figure 7—figure supplement 2-3, to explore the biological functions of AIDPS and its nine genes related to mutation, CNA, and immune infiltration in four aspects. A detailed response to this section can be found in the recommendation section of reviewer 3 in lines 644-799.

Reviewer #1 (Recommendations for the authors):Things I think would strengthen the paper:1. A better understanding of why agreement/optimization amongst a large number of combined ML methods was chosen as a way to drive feature selection. Comparing to existing methods is one thing, but there isn't really a clear comparison here to other feature selection approaches, nor much discussion of why this one was chosen. Is this meant to be a reproducible approach, for other problems, or is the point just to get the AIDPS features? Why is this approach to feature selection preferred/superior.

With advancements in high-throughput sequencing techniques and computational biology, numerous predictive signatures have been proposed according to various machine-learning approaches. However, two questions worth considering are why a particular algorithm should be selected and which solution is the optimal one. The selection of algorithms by researchers may rely largely on their own preferences and bias. Thus, to generate a consensus signature, we collected 10 prevalent algorithms and then combined them into 76 combinations. As to why we use agreement/optimization as a feature selection method, we will next answer from the following five aspects:

1. Why perform feature selection?

For clinical prediction model development, the 10 events per variable (10 EPV) rule recommends that feature variables should be included according to the number of positive events, which in general requires more than 10-fold higher number of positive events than the independent variable included (PMID: 32188600; 8970487). According to the 10 EPV rule, 1280 samples from our ten cohorts contained 772 positive events (OS of 1), which means that the independent variables included in our study at the beginning should be less than 77. However, given that there are 15,288 intersection genes in the ten cohorts, we need to conduct the first stage of variable screening for the initial covariates that are severely mismatched with the sample size. There are two commonly used variable screening methods: one is differential expression analysis of tumor and normal samples, and it is believed that differentially expressed variables may play a crucial role in the tumorigenesis and progression (e.g., PMID: 33028695; 32754277; 33391469; 34108619), and the other is univariate Cox regression analysis of survival variables, and variables with independent prognostic significance are selected for subsequent analysis (e.g., PMID: 35145098; 34922323; 33302293; 31350847; 32201203; 33038770) ; and even some studies combine the two methods (e.g., PMID: 32719094; 34237133). Because the ten cohorts lacked normal samples (there were only 4 normal samples in the TCGA-PAAD and even no normal samples in the PACA-AU-Array used for modeling), univariate Cox regression analysis was used for variable screening in this study.

2. Why choose consistent prognostic genes (agreement)?

It is well-known that the widespread tumor heterogeneity caused by different genetic background, pathological type, differentiation state and even living environment poses a great challenge to treatment and is also an urgent problem to be solved in precision medicine. Because of spatiotemporal heterogeneity, the same gene may have different prognostic performance in different cohorts, for example, A1BG was a protective factor in the TCGA-PAAD and GSE79668, and had no significant prognostic value in most cohorts, but it was indeed a risk factor in the GSE71729. Therefore, in order to obtain more stable prognostic genes, drawing on previous experience (PMID: 35145098; 34922323; 34394098), we obtained 32 consensus prognostic genes in the first stage for subsequent machine-learning modeling according to the screening criteria of consistent prognostic significance in at least 8/10 cohorts.

3. Why build machine-learning algorithm combinations?

Based on the 32 consensus prognostic genes obtained from the appeal step, we selected the PACA-AU-Array with larger sample size as the training set to construct the consensus model. However, according to the 10 EPV rule, the PACA-AU-Array (267 samples, of which 161 positive events) allowed up to 16 variables to be included for subsequent model construction. To solve this problem, among the ten machine-learning algorithms we collected, RSF, LASSO, CoxBoost and Stepwise Cox have the function of dimensionality reduction and variable screening. Therefore, we combined them with other algorithms to make the model more simplified and more conducive to clinical translation.

4. Why choose the largest average C-index (optimization)?

The pros and cons of a model lies in whether it can maintain good predictive ability in the validation sets, that is, it has strong generalization ability. In existing studies, the concordance index (C-index, PMID: 22532584; 28068175; 28687838) and the time-dependent ROC curve (timeROC, PMID: 26813360; 33667676; 34157487) are the most common and widely recognized indicators for evaluating the predictive performance of machine-learning survival models. Among them, C-index can be used to judge the accuracy and discrimination ability for global prediction, and the performance is relatively stable. While the area under the timeROC curve (AUC) depends on the time node, reflecting the prediction accuracy at a certain moment, and the overall performance is not very stable. Therefore, we chose C-index to evaluate the predictive performance of the 76 algorithm combinations.

In addition, the aim of our study was to obtain a predictive model with strong generalization ability that could maintain excellent performance in each validation cohort and have a good likelihood of clinical translation. Some models, such as random survival forest (RSF) model, performed extremely well on the PACA-AU-Array training set (C-index = 0.914), but it performed very poorly on the nine validation sets (C-index: TCGA-PAAD = 0.57, and PACA-CA-Seq = 0.61). This is mainly because some algorithms are easy to overfit in the training set. In this situation, we thus cannot say that RSF is the optimal model. Therefore, in order to avoid the impact of model overfitting as much as possible, following the previous workflow (PMID: 35145098; 34922323), we chose the largest average C-index of nine validation sets as our feature selection method.

5. From the results, how is the effect of our feature selection approach?

In summary, based on the feature selection criteria of consistent prognostic value (agreement) and the largest average C-index (optimization), we combined CoxBoost and Survival-SVM to construct a 9-gene AIDPS. In nine validation sets, in addition to owning an average C-index of 0.675, AIDPS also exhibited robust and superior predictive capability (all AUCs >0.65) in another recognized indicators, timeROC, and outperformed the existing 86 gene expression signatures. Meanwhile, the composition of 9-gene AIDPS is relatively simple, and it is more likely to construct a multigene panel for clinical translation. In addition, as recommended by reviewer 2, we randomly split the Meta-Cohort into 22 validation cohorts (from Cohort_100 to Cohort_1150) and again thoroughly collected three external test cohorts, including GSE21501 (n = 102), GSE57495 (n = 63) and GSE71729 (n = 125). The univariate Cox regression analysis, Kaplan-Meier survival curves, AUCs of timeROC curves, C-index, and calibration curves consistently demonstrated robust and good predictive performance of AIDPS (Figure 3—figure supplement 4-5). Therefore, based on the above results, we can preliminarily illustrate that AIDPS generated by our machine-learning survival framework has great potential for clinical application in PACA patients with insidious onset and poor prognosis.

2. Right now the current pipeline seems highly unwieldy because of all the models considered. Is this meant to be a general and reproducible approach? if so, the authors need to demonstrate more clearly that this same approach can be done by others with ease.

Thank you for your questions. In existing predictive models, researchers mostly chose the modelling algorithms based on their knowledge limitations and preferences, and it is worth thinking about which method is more suitable or which solution is optimal. Therefore, to construct a consensus model with robust predictive performance and stronger generalization ability and extrapolation possibilities, we collected ten classical machine-learning algorithms, among which RSF, LASSO, CoxBoost and Stepwise Cox have the functions of dimensionality reducing and variable screening. Given that our training set has only 161 positive events and a maximum of 16 variables can be accommodated for subsequent model construction according to the 10 EPV rule, we combined these four algorithms with other algorithms to generate 76 combinations to make the model more simplified and more conductive to clinical translation. This does make our pipeline seem unwieldy, but at present this method has better performance and application prospects, and the same pipeline has also been applied to the following high-level studies:

(1) In February 2022, our team published an article entitled: “Machine learning-based integration develops an immune-derived lncRNA signature for improving outcomes in colorectal cancer” in Nature Communications (PMID: 35145098). In this study, based on 43 immune-related lncRNAs with consistent prognostic significance obtained by univariate Cox regression analysis, we combined ten machine-learning algorithms into 101 algorithm combinations and demonstrated that the model constructed by Lasso combined with stepwise Cox (direction = both) possessed the best predictive performance in a training set and six validation sets (average C-index of 0.696). In addition, we confirmed that the high-risk group was sensitive to fluorouracil-based adjuvant chemotherapy, whereas the low-risk group showed more abundant immune cells infiltration and was more sensitive to pembrolizumab.

(2) In January 2022, our team published another article entitled: “Integrative analysis from multi-center studies identities a consensus machine learning-derived lncRNA signature for stage II/III colorectal cancer” in EBioMedicine (PMID: 34922323). In this study, based on 27 stable recurrence-related lncRNAs, we combined ten machine-learning algorithms into 76 algorithm combinations and demonstrated that CMDLncS generated by Lasso combined with stepwise Cox could accurately detect recurrence in patients with II/III colorectal cancer (average C-index of 0.777). In addition, patients in the high CMDLncS group were likely to benefit from fluorouracil-based adjuvant chemotherapy, whereas the low CMDLncS group showed greater sensitivity to bevacizumab.

(3) Recently, Nan Zhang et al., from Xiangya Hospital of Central South University, published the other article entitled “Machine learning-based identification of tumor-infiltrating immune cell-associated lncRNAs for improving outcomes and immunotherapy responses in patients with low-grade glioma” in Theranostics (PMID: 35966587). The authors identified 136 tumor-infiltrating immune cell-associated lncRNA (TIIClncRNA) of low-grade glioma (LGG) by comprehensive analyzing the sequencing data of purified immune cells, LGG cell lines, and bulk LGG tissues. After that, based on 46 prognostic TIIClncRNAs obtained by univariate Cox regression analysis, the authors combined ten machine-learning algorithms into 101 algorithm combinations, and picked out TIIClnc signature that derived by combining RSF and CoxBoost algorithms using the largest mean C-index (0.744) in the TCGA training set, Xiangya in-house validating set, and two external validating sets. In addition, the authors found that TIIClnc signature was significantly associated with the expression of CD8, PD-1, and PD-L1, as well as immune characteristics such as microsatellite instability, tumor mutation burden, interferon γ. Results from multiple datasets demonstrate that TIIClnc signature accurately predicts superior immunotherapy response across cancer types.

In addition, the complete flow and essential scripts of our pipeline has also been published in the GitHub website (https://github.com/Zaoqu-Liu/IRLS). To our knowledge, some people have been applying our pipeline in solid tumors such as melanoma, bladder cancer, hepatocellular carcinoma, gastric cancer and have achieved good results. For PACA, our AIDPS model has also been uploaded to the corresponding catalogue (https://github.com/Zaoqu-Liu/AIDPS), and the researchers entered the expression matrix of these nine AIDPS genes to obtain patient-specific risk score. Thanks again for your questions, which let us think again about the extensibility and generalizability of our pipeline.

3. Everything in the subsequent analyses comparing high and low AIDPS groups hinges on the assumed accuracy and generality of the model. Once you have 9 genes, though, you should be able to study those features using a range of methods, some of which are picked to have high interpretability. I am sorry if I missed this, but at the moment it is not clear to me what best an ML method can do is using the 9 genes, and whether there is an interpretable method that works well but might imply a non-trivial relationship amongst the genes that matters to the outcome. This seems more informative than studying the statistics of high and low AIDPS groups

Thanks for your questions and suggestions. As you said, once we have these nine AIDPS genes, we can conduct a series of studies. Therefore, we supplemented Figure 7—figure supplement 1-3 to further explore the relationship between AIDPS and nine AIDPS genes and prognosis, immune cells infiltration and immune checkpoint molecules (ICMs) expression. Next, we will explain our findings in following four sections:

1. Prognostic performance of AIDPS and nine AIDPS genes

Our previous results consistently indicated that AIDPS could accurately predict the prognosis of PACA patients (Figure 3-4 and Figure 3—figure supplement 1-3). According to your advice, based on large multi-center data from the training set, nine validation sets, and three newly collected external test sets, we performed an integrated analysis of survival variables using AIDPS and nine AIDPS genes as continuous variables. As shown in Figure 7—figure supplement 1A, AIDPS was an independent protective factor in all cohorts (among them, there were only 26 positive and 2 negative events in the PACA_AU_Seq, although there was a corresponding trend, it did not reach statistical significance). Correspondingly, because the 32 consensus prognostic genes used to construct AIDPS with consistent prognostic value in at least 8/10 cohorts, nine AIDPS genes had relatively consistent prognostic performance in the ten cohorts (Figure 7—figure supplement 1B-J). However, in the three external test sets, GSE21501, GSE57495 and GSE71729, the performance of nine AIDPS genes was hardly satisfactory. Overall, compared with the nine independent AIDPS genes, our AIDPS constructed by CoxBoost and Survival-SVM brings significant performance improvement, and its simple composition and robust performance also have greater possibility of clinical translation and extrapolation.

2. Correlation between AIDPS and its nine genes at expression level

To describe the biological relevance between AIDPS and its nine genes, we performed Pearson correlation analysis referring to previous studies (PMID: 32853987; 32201203). As shown in Figure 7—figure supplement 2A, SELENBP1 and PLCB4 had strong positive correlations with AIDPS (R = 0.46 and 0.57), while DCBLD2, PRR11, UNC13D, EREG, ADM, CDCA4, and TGM2 had very strong negative correlations with AIDPS (R from -0.57 to -0.75) in the whole TCGA-PAAD. Moreover, there was a positive correlation between the two genes positively correlated with AIDPS (R = 0.23). Similarly, there were also positive correlation between the seven genes negatively correlated with AIDPS (R from 0.19 to 0.59), while there were moderate negative correlation between the two genes positively correlated AIDPS and the seven genes negatively correlated AIDPS (R from 0 to -0.55). Based on the rationale of weighted gene co-expression network analysis (WGCNA, PMID: 19114008): genes with similar expression patterns may be co-regulated, functionally correlated or in the same pathway. The strong correlation within the nine genes explains the rationality of our AIDPS from the perspective of biological function to some extent.

In addition, to investigate the changes of the nine AIDPS genes with AIDPS changes, we performed Pearson correlation analysis again in the high and low AIDPS TCGA-PAAD, respectively, and the low AIDPS group exhibited a consistent trend with the whole TCGA-PAAD (Figure 7—figure supplement 2C). Interestingly, the overall correlation was lower in the high AIDPS group and numerous genes showed different or even opposite trends to the whole and the low AIDPS group (e.g., SELENBP1 changed from positive to negative correlation with AIDPS and PLCB4), suggesting that AIDPS may have stronger biological relevance and potential function in PACA patients with low AIDPS (Figure 7—figure supplement 2B).

3. Relationship between nine AIDPS genes and immune molecules and immune cells

Given the obvious impact of AIDPS in the TME and its guiding significance for immunotherapy, we compared the relationship of AIDPS and its nine genes with 27 ICMs and 28 immune cells in the whole, high AIDPS, and low AIDPS TCGA-PAAD. The results again corroborate our previous finding that the low AIDPS group had higher immune cells infiltration and ICMs expression (Figure 7). In the whole TCGA-PAAD, AIDPS showed a prominent negative correlation with ICMs and immune cells, and SELENBP1 and PLCB4, which were positively correlated with AIDPS, also exhibited a corresponding trend. However, DCBLD2, PRR11, UNC13D, EREG, and TGM2, which were negatively correlated with AIDPS, had a significant positive correlation with ICMs and immune cells (Figure 7—figure supplement 2D and G). The high AIDPS group also showed the similar results as the whole cohort (Figure 7—figure supplement 2E and H). Interestingly, in the low AIDPS group, AIDPS showed an observably positive correlation with ICMs and immune cells different from the whole and high AIDPS cohort. Meanwhile, PLCB4, which was positively correlated with AIDPS, showed a similar trend, while DCBLD2, PRR11, UNC13D, EREG, ADM, CDCA4 and TGM2, which were negatively correlated with AIDPS, were negatively correlated with ICMs and immune cells (Figure 7—figure supplement 2F and I).

4. Current research progress on these nine AIDPS genes

After careful literature research, we found that SELENBP1 that owned lower expression in the low AIDPS group with poor prognosis was associated with better tumor prognosis, and its decreased expression led to poor prognosis of many tumors such as lung adenocarcinoma. Meanwhile, DCBLD2, PRR11, EREG, ADM and TGM2, which were overexpressed in the low AIDPS group, played a vital role in the proliferation, invasion and metastasis of many tumors such as PACA, hepatocellular carcinoma and colorectal cancer, and lead to their chemotherapy resistance and were potential targets for enhanced efficacy. However, the functional roles and mechanisms of UNC13D and CDCA4 in tumors are not well understood, and more high-quality studies are needed in the future. These findings are consistent with our study. The specific functional characteristics of these nine AIDPS genes are summarized as follows:

(1) SELENBP1, encoding a novel human methanethiol oxidase, whose expression is significantly decreased in the progression of Barrett’s esophagus to adenocarcinoma and can affect chemotherapy sensitivity (PMID: 20332323). This is consistent with our results that SELENBP1 expression was decreased in the low AIDPS group with worse prognosis (Figure 7—figure supplement 3).

(2) PLCB4 encodes a phospholipase, and activating PLCB4 mutation was recently identified in primary leptomeningeal melanocytic tumors (PMID: 28499758).

(3) DCBLD2 is upregulated in glioblastoma and head and neck cancer, and phosphorylation of DCBLD2 Y750 activates AKT pathway which in turn enhances EGFR-driven tumorigenesis (PMID: 25061874). DCBLD2 has also been found to play a crucial role in the invasion, progression, and metastasis of neuroendocrine tumors (PMID: 18827820). These results are consistent with an increased expression of DCBLD2 in the low AIDPS group with a dismal prognosis.

(4) Proline-rich protein 11 (PRR11) is a novel tumor-related gene, which has been found can promote breast cancer growth and anti-estrogen resistance through the PI3K pathway (PMID: 33127913). In addition, it has also been demonstrated that PRR11 affects autophagy and promotes proliferation in non-small cell lung cancer (NSCLC) through the Akt/mTOR pathway (PMID: 30258945; 35005120), and recent studies have also found that PRR11 can promote the tumorigenesis and progression of renal clear cell carcinoma (ccRCC) by regulating E2F1 stability (PMID: 34499617). Overall, these results are consistent with higher expression of PRR11 in the low AIDPS group with a worse prognosis.

(5) UNC13D mutation play an essential role in familial hemophagocytic lymphohistiocytosis (FHL), but its role in tumors remains unknown and needs to be further explored (PMID: 21931115; 24842371; 25564401; 24882743).

(6) EREG, as an EGFR ligand epithelial regulatory protein, plays a fundamental role in the growth and proliferation of breast and colorectal cancers (PMID: 26215578; 23549083) and is associated with tamoxifen resistance in breast cancer (PMID: 30967627) and efficacy of panitumumab and adjuvant chemotherapy in colorectal cancer (PMID: 26869404; 25272957).

(7) Adrenomedullin (ADM) encodes a pre-prohormone that is cleaved into two bioactive peptides, which can dilate blood vessels, regulate hormone secretion, and promoting angiogenesis. Many studies have confirmed that ADM plays a major role in the proliferation, invasion, angiogenesis, and metastasis of PACA, hepatocellular carcinoma, prostate cancer, melanoma, ovarian cancer, and endometrial cancer and may become a novel therapeutic target (PMID: 22960655; 17363587; 18657357; 24100627; 21994414; 22400488; 11420706).

(8) In addition, currently one of the few studies support the oncogenic role of cell division cycle associated protein 4 (CDCA4) in tumors and nuclear factors induced as E2F transcription factor families that regulate E2F-dependent transcriptional activation and cell proliferation, but their potential mechanism, especially effect on PACA tumorigenesis and progression require further investigation (PMID: 35251007; 16984923).

(9) Transglutaminase 2 (TGM2), a promoter of stemness and radiotherapy resistance in glioma (PMID: 33483373), has been confirmed in many studies to promote colorectal cancer progression via P53 and Wnt/β-catenin signaling pathways (PMID: 34103685; 31570702), as well as is associated with tumor-promoting inflammation in gastric cancer (PMID: 32467608). These results are consistent with the TGM2 expression dramatically increased in the low AIDPS group with worse prognosis, suggesting that TGM2 may play an important role in the development and progression of PACA.

In summary, our study demonstrated that AIDPS outperformed nine AIDPS genes alone in prediction performance, so we chose AIDPS instead of the nine genes alone for subsequent analysis. The biological relevance studies exhibited that AIDPS and its nine genes had inconsistent or even opposite performances in the high and low AIDPS groups, suggesting that the high and low AIDPS groups owned relatively strong biological heterogeneity. In addition, our literature survey found that these nine AIDPS genes were rarely studied in tumors, and most of these focus on their roles in biological phenotypes such as tumorigenesis and progression, and few studies investigated their roles and guiding significance in PACA immune microenvironment and treatment. Meanwhile, our study showed that the high and low AIDPS groups had observably different prognosis, functional characteristics, immune infiltration, genomic variant landscape, and treatment response. Moreover, stratifying PACA patients into high and low AIDPS groups for individualized treatment and clinical management is also in line with the concept of precise treatment. Therefore, in this study, we focused on the high and low AIDPS groups. Relevant contents have been added and marked in red in the corresponding section of the revised manuscript. Thanks again for your questions and suggestions, and we are sure that your suggestions have enriched and refined our research.

Reviewer #2 (Recommendations for the authors):To rescue their claims one could suggest to the authors the following simple but clean and fair test. The authors could repeat their complete pipeline after randomizing within each of their ten data sets the clinical outcomes (i.e. shuffle the time-to-event outcomes). If that exercise is found to make the signal disappear (i.e. the separation of KM curves, and the nontrivial AUROC values), then their present results should be studied further as possibly relevant. If, on the contrary, one would obtain again a signature with comparable prognostic features, then one can be sure that what is reported here is largely a manifestation of overfitting due to dimension mismatch, and hence of no interest.

Thank you for your advice. During the construction of AIDPS, in order to eliminate the heterogeneity between different cohorts as much as possible and obtain relatively stable prognostic genes in PACA, we performed univariate Cox regression analysis and identified 32 consensus prognostic genes according to the following criteria of consistent prognostic significance in at least 8/10 cohorts. Afterwards, we selected the PACA-AU-Array with larger sample size as the training set and validated the model using the remaining nine cohorts and Meta-Cohort, and confirmed that AIDPS has robust and good predictive performance. In doing so, as you are concerned, there is indeed the possibility of applying validation sets to train the model and resulting in a lack of rigor in the validation results. But we did this to make more adequate use of the 10 PACA cohorts to obtain a robust model. In a sense, if we also use the genes of a specific biological pathway for modeling according to popular practice, then our model will be like the collected 86 signatures and is hard to generalize in other cohorts or populations, and unable to further clinical translation and serve the clinical management and precision treatment of PACA patients. To rescue our claims, as you recommended, we supplemented the contents of Figure 3—figure supplement 4-5 to further test the performance of AIDPS. The details are as follows:

1. Performance of AIDPS in randomly split cohorts

Following your recommendations, we used the sample function of R software to perform non-return random sampling from Meta-Cohort synthesized from 10 PACA cohorts, and 100, 150, 200, 250, 300, 350, 400, 450, 500, 550, 600, 650, 700, 750, 800, 850, 900, 950, 1000, 1050, 1100, and 1150 samples were successively selected to form a new cohort. Kaplan-Meier survival analysis, univariate Cox regression analysis, timeROC, and C-index were used to measure the performance of AIDPS in these cohorts. The Figure 3—figure supplement 4 is presented from left to right as follows: the results of Kaplan-Meier survival analysis showed that the survival curves of patients in the high and low AIDPS groups were remarkably separated (curves not shown; Log-rank P = 0.0019 in the Cohort_100, and <0.0001 in the other cohorts); univariate Cox regression analysis also exhibited that AIDPS was an independent protective factor in these cohorts (P <0.001 in the Cohort_100 and <0.0001 in the other cohorts); the timeROC curves for 1-, 2-, 3-, 4-, and 5-years OS also showed that AIDPS owned AUCs greater than 0.7 in the vast majority of cohorts, and at least greater than 0.65 in the relatively poor Cohort_100, Cohort_200, and Cohort_300; plus C-index greater than 0.65 in all cohorts, suggesting that our AIDPS can accurately predict OS in PACA patients.

2. Performance of AIDPS in three additional independent test cohorts

Furthermore, to more objectively test the performance of AIDPS, in addition to the training and validation sets mentioned earlier, we again screened and searched for three external test cohorts, GSE21501 (n = 102), GSE57495 (n = 63), and GSE71729 (n = 125). The results of univariate Cox regression analysis showed that AIDPS was an independent protective factor in all three cohorts (Figure 3—figure supplement 5A). Kaplan-Meier survival analysis also exhibited that patients in the low AIDPS group owned a significantly shorter OS (Figure 3—figure supplement 5B-D, Log-rank P = 0.0014 in GSE21501, 0.00045 in GSE57495, 0.00011 in GSE71729). The AUCs of 1-, 2-, and 3-year OS were 0.677, 0.681, and 0.761 in the GSE21501; 0.682, 0.728, and 0.747 in the GSE57495; 0.676, 0.693, and 0.714 in the GSE71729 (Figure 3—figure supplement 5E-G). In addition, the calibration curves of these three external test cohorts also confirmed the good predictive performance of AIDPS (Figure 3—figure supplement 5H-J).

Overall, the results of the above randomized split validation cohorts and three external test cohorts consistently confirmed that AIDPS can robustly predict the prognosis of PACA patients. Thanks again for your suggestions, and we are sure that your suggestions makes our verification more rigorous, reliable and meaningful.

Reviewer #3 (Recommendation for the authors):From the biological point of view, currently, there was little explanation of the biological relevance of the 9 genes in relation to the mutational load/copy number changes/ immunological infiltration found in low AIDPS patients. Please discuss these in sufficient detail.

Thank you for your recommendations. As you mentioned, our study explored the genomic landscape such as mutation and copy number alteration (CNA) of the 9-gene AIDPS and their impact on the tumor microenvironment (TME), focusing on its guiding value for decision-making and clinical management in PACA patients, while ignoring the biological functions of these nine genes as independent individuals. To compensate for this defect, we supplemented the correlation analysis between AIDPS and nine AIDPS genes in the whole, high AIDPS, and low AIDPS TCGA-PAAD. Considering the outstanding different TME between the high and low AIDPS groups, we added a correlation analysis of AIDPS and its nine genes with 27 immune checkpoint molecules (ICMs) and 28 immune cells obtained by ssGSEA in the whole, high AIDPS, and low AIDPS TCGA-PAAD. In addition, to describe the overall relationship between AIDPS and its nine genes and mutation/CNA in PACA patients, we added expression information of these genes in the process of Figure 6A to generate Figure 7—figure supplement 3. Next, we will explore the biological relevance between AIDPS and its nine genes with mutation, CNA, and immune infiltration from the following four aspects:

1. Correlation between AIDPS and its nine genes at expression level

To describe the biological relevance between AIDPS and its nine genes, we first performed Pearson correlation analysis referring to previous studies (PMID: 32853987; 32201203). As shown in Figure 7—figure supplement 2A, SELENBP1 and PLCB4 had strong positive correlations with AIDPS (R = 0.46 and 0.57), while DCBLD2, PRR11, UNC13D, EREG, ADM, CDCA4, and TGM2 had very strong negative correlations with AIDPS (R from -0.57 to -0.75) in the whole TCGA-PAAD. Moreover, there was a positive correlation between the two genes positively correlated with AIDPS (R = 0.23). Similarly, there were also positive correlation between the seven genes negatively correlated with AIDPS (R from 0.19 to 0.59), while there were moderate negative correlation between the two genes positively correlated AIDPS and the seven genes negatively correlated AIDPS (R from 0 to -0.55). Based on the rationale of weighted gene co-expression network analysis (WGCNA, PMID: 19114008): genes with similar expression patterns may be co-regulated, functionally correlated or in the same pathway. The strong correlation within the nine genes explains the rationality of our AIDPS from the perspective of biological function to some extent.

In addition, to investigate the changes of the nine AIDPS genes with AIDPS changes, we performed Pearson correlation analysis again in the high and low AIDPS TCGA-PAAD, respectively, and the low AIDPS group exhibited a consistent trend with the whole TCGA-PAAD (Figure 7—figure supplement 2C). Interestingly, the overall correlation was lower in the high AIDPS group and numerous genes showed different or even opposite trends to the whole and the low AIDPS group (e.g., SELENBP1 changed from positive to negative correlation with AIDPS and PLCB4), suggesting that AIDPS may have stronger biological relevance and potential function in PACA patients with low AIDPS (Figure 7—figure supplement 2B).

2. Relationship between nine AIDPS genes and immune molecules and immune cells

Given the obvious impact of AIDPS in the TME and its guiding significance for immunotherapy, we compared the relationship of AIDPS and its nine genes with 27 ICMs and 28 immune cells in the whole, high AIDPS, and low AIDPS TCGA-PAAD. The results again corroborate our previous finding that the low AIDPS group had superior immune cells infiltration and ICMs expression (Figure 7). In the whole TCGA-PAAD, AIDPS showed a prominent negative correlation with ICMs and immune cells, and SELENBP1 and PLCB4, which were positively correlated with AIDPS, also exhibited a corresponding trend. However, DCBLD2, PRR11, UNC13D, EREG, and TGM2, which were negatively correlated with AIDPS, had a significant positive correlation with ICMs and immune cells (Figure 7—figure supplement 2D and G). The high AIDPS group also showed the similar results as the whole cohort (Figure 7—figure supplement 2E and H). Interestingly, the tread in the low AIDPS group was contrary, AIDPS and PLCB4 were observably positively correlated with ICMs and immune cells. Meanwhile, DCBLD2, PRR11, UNC13D, EREG, ADM, CDCA4 and TGM2, which were negatively correlated with AIDPS, were negatively correlated with ICMs and immune cells (Figure 7—figure supplement 2F and I).

The disparate result of AIDPS and its nine genes in the high and low AIDPS groups indicated that these AIDPS genes may be closely related to the immune regulatory pathway, which may play a crucial role in anti-tumor immune and immunotherapy. This also provides a rationale to some extent that AIDPS can accurately predict immunotherapy response in PACA patients.

3. Relevance of nine AIDPS genes with mutation and copy number alteration

Our previous study found that the low AIDPS group owned outstandingly higher genomic alterations such as tumor mutation burden (TMB) and CNA load, which could provide more neoantigens for anti-tumor immune and immunotherapy and lead to better immunotherapeutic efficacy (Figure 6A-D). As shown in Figure 7—figure supplement 3, the proportion of top 15 gene mutation and CNA loci (frequency >20%) were obviously higher in the low AIDPS group. In addition, we also found that SELENBP1 and PLCB4, which were positively correlated with AIDPS, were significantly lower in the low AIDPS group, suggesting that they were significantly negatively correlated with TMB and CNA burden. Correspondingly, DCBLD2, PRR11, UNC13D, EREG, ADM, CDCA4, and TGM2, which were negatively correlated with AIDPS, was remarkably increased in the low AIDPS group, hinting they were significantly positively correlated with TMB and CNA burden (e.g., the low AIDPS group possessed higher TP53 mutation and 8q24.21 amplification, as well as higher DCBLD2 expression).

4. Current research progress on these nine AIDPS genes

After careful literature research, we found that the nine AIDPS genes were rarely studied in tumors, most of which focused on their roles in biological phenotypes such as tumorigenesis and progression, and few studies investigated their roles and guiding significance in PACA immune microenvironment and treatment. Specifically, we found that SELENBP1 owned lower expression in the low AIDPS group with poor prognosis was associated with better tumor prognosis, and its decreased expression resulted in poor prognosis of many tumors such as lung adenocarcinoma. Meanwhile, DCBLD2, PRR11, EREG, ADM and TGM2, which were overexpressed in the low AIDPS group, played an essential role in the proliferation, invasion and metastasis of many tumors such as PACA, hepatocellular carcinoma and colorectal cancer, and lead to their chemotherapy resistance and were potential targets for enhanced efficacy. However, the functional roles and mechanisms of UNC13D and CDCA4 in tumors are not well understood, and more high-quality studies are needed in the future. These findings are consistent with our study, and the specific functional characteristics are summarized as follows:

(1) SELENBP1, encoding a novel human methanethiol oxidase, whose expression is significantly decreased in the progression of Barrett’s esophagus to adenocarcinoma and can affect chemotherapy sensitivity (PMID: 20332323). This is consistent with our results that SELENBP1 expression was decreased in the low AIDPS group with worse prognosis (Figure 7—figure supplement 3).

(2) PLCB4 encodes a phospholipase, and activating PLCB4 mutation was recently identified in primary leptomeningeal melanocytic tumors (PMID: 28499758).

(3) DCBLD2 is upregulated in glioblastoma and head and neck cancer, and phosphorylation of DCBLD2 Y750 activates AKT pathway which in turn enhances EGFR-driven tumorigenesis (PMID: 25061874). DCBLD2 has also been found to play a crucial role in the invasion, progression, and metastasis of neuroendocrine tumors (PMID: 18827820). These results are consistent with an increased expression of DCBLD2 in the low AIDPS group with a dismal prognosis.

(4) Proline-rich protein 11 (PRR11) is a novel tumor-related gene, which has been found can promote breast cancer growth and anti-estrogen resistance through the PI3K pathway (PMID: 33127913). In addition, it has also been demonstrated that PRR11 affects autophagy and promotes proliferation in non-small cell lung cancer (NSCLC) through the Akt/mTOR pathway (PMID: 30258945; 35005120), and recent studies have also found that PRR11 can promote the tumorigenesis and progression of renal clear cell carcinoma (ccRCC) by regulating E2F1 stability (PMID: 34499617). Overall, these results are consistent with higher expression of PRR11 in the low AIDPS group with a worse prognosis.

(5) UNC13D mutation play an essential role in familial hemophagocytic lymphohistiocytosis (FHL), but its role in tumors remains unknown and needs to be further explored (PMID: 21931115; 24842371; 25564401; 24882743).

(6) EREG, as an EGFR ligand epithelial regulatory protein, plays a fundamental role in the growth and proliferation of breast and colorectal cancers (PMID: 26215578; 23549083) and is associated with tamoxifen resistance in breast cancer (PMID: 30967627) and efficacy of panitumumab and adjuvant chemotherapy in colorectal cancer (PMID: 26869404; 25272957).

(7) Adrenomedullin (ADM) encodes a pre-prohormone that is cleaved into two bioactive peptides, which can dilate blood vessels, regulate hormone secretion, and promoting angiogenesis. Many studies have confirmed that ADM plays a major role in the proliferation, invasion, angiogenesis, and metastasis of PACA, hepatocellular carcinoma, prostate cancer, melanoma, ovarian cancer, and endometrial cancer and may become a novel therapeutic target (PMID: 22960655; 17363587; 18657357; 24100627; 21994414; 22400488; 11420706).

(8) In addition, currently one of the few studies support the oncogenic role of cell division cycle associated protein 4 (CDCA4) in tumors and nuclear factors induced as E2F transcription factor families that regulate E2F-dependent transcriptional activation and cell proliferation, but their potential mechanism, especially effect on PACA tumorigenesis and progression require further investigation (PMID: 35251007; 16984923).

(9) Transglutaminase 2 (TGM2), a promoter of stemness and radiotherapy resistance in glioma (PMID: 33483373), has been confirmed in many studies to promote colorectal cancer progression via P53 and Wnt/β-catenin signaling pathways (PMID: 34103685; 31570702), as well as is associated with tumor-promoting inflammation in gastric cancer (PMID: 32467608). These results are consistent with the TGM2 expression dramatically increased in the low AIDPS group with worse prognosis, suggesting that TGM2 may play an important role in the development and progression of PACA.

In conclusion, previous studies and our analysis results consistently indicate that the nine AIDPS genes are correlated with mutation, CNA, and TME in PACA. This also essentially supports the important impact of AIDPS on the prognosis and immune microenvironment, as well as its role in the clinical management and precision treatment of PACA. The relevant contents have been added and marked in red in the corresponding section of the revised manuscript. Thanks again for your valuable comments, and we are sure that your hard work has made our study more rigorous and meaningful.

[Editors' note: further revisions were suggested prior to acceptance, as described below.]

The manuscript has been improved but there are some remaining issues that need to be addressed, as outlined below:1. The improvements to figure 7 (in the supplementary figures linked to Figure 7) meant to address Reviewer #1's request seem to all focus on treating the aidps features separately. Please show the accuracy when these aidps features are considered together as a nine dimensional feature vector in one model.

Thank you and Reviewer #1 for your reminder. As you mentioned, in this revision, we supplemented the content of Figure 4—figure supplement 1D, focusing on exploring the performance of this 9 AIDPS genes in other machine-learning algorithms. The results exhibited that our AIDPS model constructed by survival-SVM is the best choice for 9 AIDPS genes obtained after dimensionality reduction by CoxBoost from 32 consensus prognostic genes, which is also consistent with our previous pipeline results (Figure 2A). Specific responses are presented in the recommendations section of Reviewer #1 in lines 69-88.

2. For reviewer #2, please remove the newly added material called `performance of AIDPS in randomly split cohorts', as this adds nothing relevant (as it s again testing on discovery data). Please take out all mentioning of `validation' or `validated' from the manuscript, other than when discussing the tests on the three new data sets GSE21501, GSE57495, GSE71729; only the latter can be called validation sets. Please compare the performance of existing signatures to that of the authors' own signature on the three new data sets GSE21501, GSE57495, GSE71729.The main reason for this suggestion is that the revised manuscript still tests their own signature on discovery data but that of the competitors on unseen data, which is not a fair comparison. Unless the authors remedy this they should (i) remove all mentioning of superior performance of their signature (since the comparison did not involve a level paying field) and (ii) remove all mentioning of `validation' other than in relation to the three new data sets.

Thank you and Reviewer #2 for your suggestions. Following the Option B of Reviewer #2’s recommendations, we have removed all contents related to the “randomly split cohorts” added in the first revision. In addition, in our latest manuscript, we refer to the PACA-AU-Array as the training cohort, the remaining nine cohorts involved in screening consensus prognostic genes as the testing cohorts, while the three external datasets GSE21501, GSE57495, and GSE71729 are called the validation cohorts. Moreover, in this revised manuscript, we checked our wording again and used the “validated” and “validation” only in these three validation cohorts. Furthermore, to remedy our claims, we compared the performance of our AIDPS with 86 published signatures in the three new validation cohorts GSE21501, GSE57495 and GSE71729, and again demonstrated our AIDPS stability and superior performance (Figure 4—figure supplement 1). A detailed response to this section can be found in the recommendations section of Reviewer #2 in lines 151-190.

3. Please include (4. Current research progress on these nine AIDPS genes) in the rebuttal, which is a summary of what is known about the biological function of the nine genes, as an Appendix or Supplementary information in the manuscript.

Thank you and Reviewer #3 for your suggestions, and we have placed the section “the current research progress of these nine AIDPS genes” in Appendix 1 and uploaded it with this revised manuscript.

Reviewer #1 (Recommendations for the authors):The authors have largely addressed the concerns raised in my previous review. As best I can tell, they have not yet shown how a simple ML model (such as linear regression or xgboost) performs on the same task when only fed with the 9 aidps features simultaneously. The improvements to figure 7 meant to address my request seem to all focus on treating the aidps features separately, so it useful to know they perform poorly when separate… what about when they are considered together as a nine dimensional feature vector in one model?

Thank you for your recommendations. As you mentioned, in the last revision, we confirmed that nine AIDPS genes alone do not predict the prognosis of PACA patients, ignoring their performance as a nine-dimensional feature vector. In this revision, we supplemented the content of Figure 4—figure supplement 1D, focusing on exploring the performance of this 9 AIDPS genes in other machine-learning algorithms. It is worth pointing out that since these nine AIDPS genes are already dimensionality reduced from 32 consensus prognostic genes by CoxBoost, machine-learning algorithm combinations are no longer considered here. Therefore, we used the expression files of these nine AIDPS genes in the PACA-AU-Array training cohort to construct 18 models via 10 common machine-learning algorithms (α values for elastic networks from 0.1-0.9) and interrogated their performance in the remaining 12 multi-center cohorts. The results are shown as follows:

As expected, among all 18 models, the model built by survival-SVM, namely our 9-gene AIDPS, achieved a maximum mean C-index of 0.666 in the remaining 12 multi-center cohorts. That is, for the 9 genes obtained from 32 consensus prognostic genes after dimensionality reduction by CoxBoost, these results reconfirmed our previous pipeline results (Figure 2A), and the AIDPS model constructed by survival-SVM was the best choice. Thanks again for your reminder, we are sure that your suggestions make our study more rigorous and meaningful.

Reviewer #2 (Recommendations for the authors):In my first report I put forward two points. First, that the authors used the same data sets both in the discovery stage and the validation stage. Second, that they compared the performance of competitor signatures to theirs by using these same data. This has the following consequences: 1. They could not claim that their signature had been validated, because what they called validation data were in fact the discovery data (the potential impact of overfitting is hence unclear). 2. The comparison between their signature and the existing ones was not fair (their own signature was tested on its corresponding discovery data, but the competitors' signatures on what for those signatures would be independent validation data). Therefore, all claims on their signature having been validated and on it being superior to existing signatures were not supported by the available evidence. I also suggested that they could test for overfitting impact by repeating their previous analyses step by step with outcome-randomized versions of the original data. This should make all true signals disappear and is hence a clean and fair test.In the revised version of their manuscript the authors did not carry out the requested re-analysis with outcome-randomized data. Instead they (a) sampled several new data sets from their original data (without outcome randomization), and (b) tested their signature on three independent new data sets. Exercise (a) seems pointless; of course the signature would again be able to predict, since one is testing it once more on samples from the discovery set. Exercise (b), although not what was requested by this referee, is more useful. Here the authors report again significant performance of their signature, with AUC values in the range 0.68-0.70, lending weight to the suggestion that their signature makes sense.I conclude from the authors' reply and from the new version of the paper that point 1. above has been partly addressed, but not point 2. If the authors, as it seems, do not wish to carry out the requested analysis with outcome randomization, then they have the following options:Option A;– Remove the newly added material called `performance of AIDPS in randomly split cohorts', this adds nothing relevant (see my notes above, it is again testing performance on discovery data).– Take out all mentioning of `validation' or `validated' from the manuscript, other than when discussing the tests on the three new data sets GSE21501, GSE57495, GSE71729; only the latter can be called validation sets.– Remove all performance comparisons between the new signature and the existing ones (since this comparison is presently not fair) and all conclusions based on these comparisons.Option B:– Remove the newly added material called `performance of AIDPS in randomly split cohorts', this adds nothing relevant (see my notes above, it is again testing on discovery data).– Take out all mentioning of `validation' or `validated' from the manuscript, other than when discussing the tests on the three new data sets GSE21501, GSE57495, GSE71729; only the latter can be called validation sets.– Compare the performance of existing signatures to that of their own signature on the three new data sets GSE21501, GSE57495, GSE71729; this comparison would be fair.I would personally strongly suggest option B, since otherwise the paper would in my view be too weak in terms of the amount of justified conclusions to warrant publication (but that is a decision for the editor to make). In addition, I would still hope that the authors choose to comply with my original request for application of the full pipeline to outcome-randomized discovery data. This would clarify objectively, for the authors' own benefit, the extent to which their original tests and conclusions may have been affected by overfitting.

Thank you for your thoughtful suggestions. Following the Option B, we have removed all contents related to the “randomly split cohorts” added in the first revision. In addition, in our latest manuscript, we refer to the PACA-AU-Array as the training cohort, the remaining nine cohorts involved in screening consensus prognostic genes including TCGA-PAAD, PACA-AU-Seq, PACA-CA-Seq, E-MTAB-6134, GSE62452, GSE28735, GSE78229, GSE79668, and GSE85916 cohorts as the testing cohorts, while the three external datasets GSE21501, GSE57495, and GSE71729 are called the validation cohorts. Moreover, in this revised manuscript, we checked our wording again and used the “validated” and “validation” only in these three validation cohorts. Furthermore, to remedy our claims, we compared the performance of our AIDPS with 86 published signatures in the three new validation cohorts GSE21501, GSE57495 and GSE71729, and again demonstrated our AIDPS stability and superior performance. The results are exhibited in Figure 4—figure supplement 1:

As shown in Figure 4—figure supplement 1A-C, our AIDPS has robust predictive performance, ranking fourth in the GSE57495 cohort as well as fifth in the GSE21501 and GSE71729 cohorts, which is superior to almost all existing signatures. Notably, although the 6-gene signature of Stratford JK was significantly better than AIDPS in the GSE21501 and GSE71729 cohorts, it was constructed in the GSE21501 cohort and performed very poorly in other cohorts, with C-index even less than 0.6 in the GSE57495, TCGA-PAAD, PACA-AU-Seq, et al. (Figure 4—figure supplement 1E). The 15-gene signature of Chen DT had observably superior performance in his own training cohort GSE57495, but it was unsatisfactory in the GSE21501, GSE71729, PACA-CA-Seq, and other cohorts (Figure 4—figure supplement 1F). Similarly, Kim J’s 5-gene signature performed well in the training cohort GSE71729 as well as GSE21501 and PACA-AU-Seq, but very poorly in most other cohorts such as GSE85916, GSE57495, PACA-CA-Seq, and E-MTAB-6134 (Figure 4—figure supplement 1G). In addition, as shown in Author response image 1, although the 20-gene signature of Demirkol CS, the 3-gene signature of Chen H, the 6-gene signature of Hou J, the 7-gene signature of Liu X, and the 6-gene signature of Yu X_2 were superior to AIDPS in their training cohort or a few other cohorts, only our AIDPS possessed acceptable performance in all PACA cohorts, and the vast majority of cohorts had good performance with a C-index greater than 0.65.

**Author response image 1. sa2fig1:** 

Overall, using the PACA-AU-Array as the training cohort, nine PACA cohorts as the testing cohorts, the three external datasets GSE21501, GSE57495, and GSE71729 as the validation cohorts, the results of all cohorts confirmed that our AIDPS could better predict the prognosis of PACA patients. Thank you again for your questions and we are sure that your suggestions make our validation more rigorous, credible and persuasive.

Reviewer #3 (Recommendations for the authors):The authors have addressed my comments. I find the following v useful as a summary of what is known about the biological function of the nine genes and wonder if it can be incorporated as an Appendix or Supplementary information:

Thank you for your suggestions, and we have placed the section “the current research progress of these nine AIDPS genes” in Appendix 1 and uploaded it with this revised manuscript.